# Inverse Reinforcement Learning with the Average Reward Criterion

**Feiyang Wu**
feiyangwu@gatech.edu

**Jingyang Ke**
jingyang.ke@gatech.edu

**Anqi Wu**
anqiwu@gatech.edu
School of Computational Science and Engineering
College of Computing
Georgia Institute of Technology
Atlanta, Georgia 30332

## Abstract

We study the problem of Inverse Reinforcement Learning (IRL) with an average-reward criterion. The goal is to recover an unknown policy and a reward function when the agent only has samples of states and actions from an experienced agent. Previous IRL methods assume that the expert is trained in a discounted environment, and the discount factor is known. This work alleviates this assumption by proposing an average-reward framework with efficient learning algorithms. We develop novel stochastic first-order methods to solve the IRL problem under the average-reward setting, which requires solving an Average-reward Markov Decision Process (AMDP) as a subproblem. To solve the subproblem, we develop a Stochastic Policy Mirror Descent (SPMD) method under general state and action spaces that needs $\mathcal{O}(1/\varepsilon)$ steps of gradient computation. Equipped with SPMD, we propose the Inverse Policy Mirror Descent (IPMD) method for solving the IRL problem with a $\mathcal{O}(1/\varepsilon^2)$ complexity. To the best of our knowledge, the aforementioned complexity results are new in IRL literature with the average reward criterion. Finally, we corroborate our analysis with numerical experiments using the MuJoCo benchmark and additional control tasks.

## 1 Introduction

Reinforcement Learning (RL) problems are frequently formulated as Markov Decision Processes (MDPs). The agent learns a policy to maximize the reward gained over time. However, in numerous engineering challenges, we are typically presented with a set of state-action samples from experienced agents, or *experts*, without an explicit reward signal. Inverse Reinforcement Learning (IRL) aspires to recover the expert's policy and reward function from these collected samples.

Methods like Imitation Learning [29, 15, 10] reduces the IRL problem to predicting an expert's behavior without estimating the reward signal. Such formulation is undesirable in some scenarios when we either wish to continue training or analyze the structure of the agent's reward signal, e.g., in reward discovery for animal behavior study [31, 34, 26, 14]. Furthermore, as a popular methodology in Imitation Learning, Generative Adversarial Network (GAN) [15, 42] suffers from unstable training due to mode collapse [32]. The sampling-based Bayesian IRL (BIRL) approach [8, 9, 6] treats the reward function as a posterior distribution from the observation. This approach is conceptually straightforward but suffers from slow convergence of sampling. Another line of research reformulates

the IRL as RL with general utility functions [36, 39, 12, 1]. While theoretically attractive, such a formulation does not learn reward functions.

Furthermore, inferring rewards poses an additional challenge: without any additional assumption, the same expert behavior can be explained by multiple reward signals [40]. To resolve this ambiguity, the authors propose adding the principle of *Maximum Entropy* that favors a solution with a higher likelihood of the trajectory. In the following work [41], the objective is replaced by maximizing the entropy of the policy, which better characterizes the nature of stochastic sequential decision-making. A body of work focuses on alleviating the computational burden of the nested optimization structure of under the Maximum Entropy framework. In [11], the authors skipped the reward function update stage by representing rewards using action-value functions. $f$-IRL [23] minimize the divergence of state visitation distribution for finite-horizon MDPs. More recently, in [37], the authors propose a dual formulation of the original Maximum Entropy IRL and a corresponding algorithm to solve IRL in the discounted setting.

Despite the success in practice, theoretical understanding of RL/IRL methods is lacking. For RL, specifically for Discounted Markov Decision Processes (DMDPs), analysis of policy gradient methods for solving DMDPs is catching up only until recently [2, 16, 5, 19, 18]. Meanwhile, understanding of the Average-reward Markov Decision Processes (AMDPs) remains very limited, where the policy aims to maximize the long-run average reward gained. Li et al.[20] propose stochastic first-order methods for solving AMDPs with a rate of convergence $\tilde{\mathcal{O}}(\log(\varepsilon^{-1}))$ for policy optimization, but it is constrained to finite states and actions with policy explicitly represented in tabular forms. Moreover, for IRL, the only known finite time analysis are [35, 37, 38, 21] for either solving DMDPs, or finite-horizon MDPs. There are no well-established convergence analyses for IRL with AMDPs. Moreover, prior works only focus on MDPs with finite state and action spaces, which do not fully address many real-life problems that inherently feature continuous elements (e.g., robotics [17]). Such settings further require general function approximations, the usage of which lacks theoretical understanding in recent work as well.

In summary, all previous works either do not fundamentally address our problem, which necessitates the learning of reward functions, or they miss key elements such as general state and action spaces, general function approximation, and the analysis incorporating approximation errors. Each of these facets demands substantial effort. In this paper, we focus on convergence analyses for average-award MDPs. We extend the RL algorithm [20] for AMDPs to general state and action spaces with general function approximation and develop an algorithm to solve the IRL problem with the average-reward criterion. Analyses of both algorithms are new in the literature to the best of our knowledge.

## 1.1 Main contributions

**Stochastic Policy Mirror Descent for solving AMDPs**: Our research introduces the Stochastic Policy Mirror Descent (SPMD) algorithm, a novel approach designed to tackle Average-reward Markov Decision Processes (AMDPs) that involve general state and action spaces. Leveraging a performance difference lemma, we demonstrate that the SPMD algorithm converges within $\mathcal{O}(\varepsilon^{-1})$ steps for a specific nonlinear function approximation class, and in no more than $\mathcal{O}(\varepsilon^{-2})$ steps for general function approximation.

**Inverse Policy Mirror Descent for solving Maximum Entropy IRL**: We expound a dual form of the Maximum Entropy IRL problem under the average-reward framework with the principle of Maximum Entropy. This dual problem explicitly targets the minimization of discrepancies between the expert's and agent's expected average reward. Consequently, we propose a first-order method named Inverse Policy Mirror Descent (IPMD) for effectively addressing the dual problem. This algorithm operates by partially resolving an Entropy Regularized RL problem at each iteration, which is systematically solved by employing SPMD. Drawing upon the *two-timescale* stochastic approximation analysis framework [3], we present the convergence result and establish a $\mathcal{O}(\varepsilon^{-2})$ rate of convergence.

**Numerical experiments**: Our RL and IRL methodologies have been tested against the well-known robotics manipulation benchmark, MuJoCo, as a means to substantiate our theoretical analysis. The results indicate that the proposed SPMD and IPMD algorithms generally outperform state-of-the-art algorithms. In addition, we found that the IPMD algorithm notably reduces the error in recovering the reward function.

## 2 Background and problem setting

### 2.1 Average reward Markov decision processes

An Average-reward Markov decision process is described by a tuple $\mathcal{M} := (\mathcal{S}, \mathcal{A}, \mathsf{P}, c)$, where $\mathcal{S}$ denotes the state space, $\mathcal{A}$ denotes the action space, $\mathsf{P}$ is the transition kernel, and $c$ is the cost function. At each time step, the agent takes action $a \in \mathcal{A}$ at the current state $s \in \mathcal{S}$ according to its policy $\pi : \mathcal{S} \to \mathcal{A}$. We use $\Pi$ as the space of all feasible policies. Then the agent moves to the next state $s' \in \mathcal{S}$ with probability $\mathsf{P}(s'|s, a)$, while the agent receives an instantaneous cost $c(s, a)$ (or a reward $r(s, a) = -c(s, a)$). The agent's goal is to determine a policy that minimizes the long-term cost

$$\rho^\pi(s) := \lim_{T \to \infty} \frac{1}{T} \mathbb{E}_\pi \left[ \sum_{t=0}^{T-1} (c(s_t, a_t) + h^\pi(s_t)) \big| s_0 = s \right], \tag{1}$$

where $h^\pi$ is a closed convex function with respect to the policy $\pi$, i.e., there exists $\mu_h \geq 0$ such that

$$h^\pi(s) - [h^{\pi'}(s) + \langle \nabla h^{\pi'}(s, \cdot), \pi(\cdot|s) - \pi'(\cdot|s) \rangle] \geq \mu_h D(\pi(s), \pi'(s)), \tag{2}$$

where $\nabla h^{\pi'}$ denotes the subgradient of $h$ at $\pi'$ and $D(\cdot, \cdot)$ is the Bregman distance, i.e.,

$$D(a_2, a_1) := \omega(a_1) - [\omega(a_2) + \langle \omega(a_2)', a_1 - a_2 \rangle] \geq \frac{1}{2} \|a_1 - a_2\|^2, \text{ for all } a_1, a_2 \in \mathcal{A}. \tag{3}$$

Here $\omega : \mathcal{A} \to \mathbb{R}$ is a strongly convex function with the associated norm $\| \cdot \|$ in the action space $\mathcal{A}$ and let us denote $\| \cdot \|_*$ as its dual norm. In this work we also utilize a span semi-norm [27]: $\|v\|_{sp,\infty} := \max v - \min v, \forall v \in \mathbb{R}^n$. If $h^\pi = 0$, Eq. 1 reduces to the classical unregularized AMDP. If $h^\pi(s) = \mathbb{E}_\pi[-\log \pi] =: \mathcal{H}(\pi(s))$, i.e., the (differential) entropy, Eq. 1 defines the average reward of the so-called entropy-regularized MDPs.

In this work, we consider the *ergodic* setting, for which we make the following assumption formally:

**Assumption 2.1.** For any feasible policy $\pi$, the Markov chain induced by policy $\pi$ is ergodic. The Markov chain is Harris ergodic in general state and action spaces (see [22]). The stationary distribution $\kappa^\pi$ induced by any feasible policy exists and is unique. There is some constant number $0 < \Gamma < 1$ such that $\kappa^\pi(s) \geq 1 - \Gamma$.

As a result of Assumption 2.1, for any feasible policy $\pi$, the average-reward function does not depend on the initial state (see Section 8 of [27]). Given that, one can view $\rho^\pi$ as a function of $\pi$. For a given policy we also define the *basic differential value function* (also called *bias function*; see, e.g., [27])

$$\bar{V}^\pi(s) := \mathbb{E} \left[ \sum_{t=0}^\infty c(s_t, a_t) + h^\pi(s_t) - \rho^\pi | s_0 = s, a_t \sim \pi(\cdot|s_t), s_{t+1} \sim \mathsf{P}(\cdot|s_t, a_t) \right], \tag{4}$$

and the *basic differential action-value function (or basic differential Q-function)* is defined as

$$\bar{Q}^\pi(s, a) := \mathbb{E} \left[ \sum_{t=0}^\infty c(s_t, a_t) + h^\pi(s_t) - \rho^\pi | s_0 = s, a_0 = a, a_t \sim \pi(\cdot|s_t), s_{t+1} \sim \mathsf{P}(\cdot|s_t, a_t) \right]. \tag{5}$$

Moreover, we define the sets of *differential value functions* and *differential action-value functions (differential Q-functions)* as the solution sets of the following Bellman equations, respectively,

$$\bar{V}^\pi(s) = \mathbb{E}[\bar{Q}^\pi(s, a)|a \sim \pi(\cdot|s)], \tag{6}$$

$$\bar{Q}^\pi(s, a) = c(s, a) + h^{\pi(s)}(s) - \rho^\pi(s) + \int \mathsf{P}(ds' \mid s, a) \bar{V}^\pi(s'). \tag{7}$$

Under Assumption 2.1, the solution of Eq. (6) (resp., Eq. (7)) is unique up to an additive constant. Finally, our goal in solving an AMDP is to find an optimal policy $\pi^*$ that minimizes the average cost:

$$\rho^* = \rho^{\pi^*} = \min_\pi \rho^\pi \quad \text{s.t.} \quad \pi(\cdot|s) \in \Pi, \ \forall s \in \mathcal{S}. \tag{AMDP}$$

### 2.2 Inverse Reinforcement Learning

Suppose there is a near-optimal policy, and we are given its demonstrations $\zeta := \{(s_i, a_i)\}_{i \geq 1}$. IRL aims to recover a reward function such that the estimated reward best explains the demonstrations. We consider solving the IRL problem under the Maximum Entropy framework (MaxEnt-IRL), which

aims to find a reward representation that maximizes the entropy of the corresponding policy and incorporates feature matching as a constraint. Formally, the MaxEnt-IRL problem is described as

$$\max_\pi \mathcal{H}(\pi) \coloneqq \mathbb{E}_{(s,a)\sim d^\pi(\cdot,\cdot)}[-\log \pi(a|s)] \qquad \text{(MaxEnt-IRL)}$$

$$\text{s.t. } \mathbb{E}_{(s,a)\sim d^\pi}[\varphi(s,a)] = \mathbb{E}_{(s,a)\sim d^E}[\varphi(s,a)]$$

where $d^E$ and $d^\pi$ denote the state-action distribution induced by the expert and current policy $\pi^E$ and $\pi$ respectively, and $\varphi(s,a) \in \mathbb{R}^n$ denotes the feature of a given $(s,a)$ pair. If we assume that for a given state-action pair, the cost is a linear function to its feature, i.e., $c(s,a;\theta) = \theta^T \varphi(s,a)$ for some parameter $\theta$ with the same dimension of the feature $\varphi(s,a)$, we can show that the parameter $\theta$ is the dual multiplier of the above optimization problem, as the Lagrangian function can be written as

$$\mathcal{L}(\pi,\theta) = \mathcal{H}(\pi) + \mathbb{E}_{(s,a)\sim d^E}[c(s,a;\theta)] - \mathbb{E}_{(s,a)\sim d^\pi}[c(s,a;\theta)]. \qquad (8)$$

Therefore, the dual problem is formulated as

$$\min_\theta \quad L(\theta) \coloneqq \mathbb{E}_{(s,a)\sim d^E}[c(s,a;\theta)] - \mathbb{E}_{(s,a)\sim d^\pi}[c(s,a;\theta)] \qquad \text{(Dual IRL)}$$

$$\text{s.t.} \quad \pi = \arg\max_{\pi'} \mathbb{E}_{(s,a)\sim d^{\pi'}}[-c(s,a;\theta)] + \mathcal{H}(\pi').$$

Notice that to find $\pi$, we need to solve an Entropy Regularized Reinforcement Learning problem, i.e., solving AMDP with the regularizer set to be the negative entropy, $h^\pi = -\mathcal{H}(\pi)$. To this end, we first propose a Stochastic Policy Mirror Descent (SPMD) method for solving AMDP. The solution methods are presented in section 3. Then we introduce an Inverse Policy Mirror Descent (IPMD) algorithm based on SPMD for the inverse RL problem (Dual IRL), introduced in section 4. We will see that we only need to solve the subproblem AMDP partially.

# 3 Stochastic Policy Mirror Descent for AMDPs

This section proposes an SPMD method to solve Regularized Reinforcement Learning problems for AMDPs. SPMD operates in an actor-critic fashion. In each step, the agent evaluates its current policy (critic step) and performs a policy optimization step (actor step). In this work, we assume there is a way to perform the critic step using standard methods, e.g., Temporal Difference learning using neural networks. Further discussion on implementation is included in section 3.1. We will focus on designing a novel actor step and providing its complexity analysis.

Our policy optimization algorithm is motivated by the following performance difference lemma , which characterizes the difference in objective values of two policies $\pi, \pi'$.

**Lemma 3.1.** *(Performance Difference) Assume that Assumption 2.1 holds. For any $\pi, \pi' \in \Pi$*

$$\rho^{\pi'} - \rho^\pi = \int \psi^\pi(s,\pi'(s))\kappa^{\pi'}(ds), \forall s \in \mathcal{S}, \qquad (9)$$

*where*

$$\psi^\pi(s,\pi'(s)) \coloneqq \bar{Q}^\pi(s,\pi'(s)) - \bar{V}^\pi(s) + h^{\pi'}(s) - h^\pi(s). \qquad (10)$$

Proof can be found in A.1. The above lemma shows the gradient of the objective function with respect to action $a$ is $\bar{Q}^\pi(s,\pi(s)) + h^\pi(s)$, i.e.,

$$\nabla_a \rho^\pi = \int \nabla_a(\bar{Q}^\pi(s,\pi(s)) + h^\pi(s))\kappa^\pi(ds). \qquad (11)$$

The existence of the above equation requires the differentiability of $\bar{Q}$, $h$ and locally Lipschitz continuity of $\kappa^\pi$ [see 24, 25]. Note that $\psi$ can be seen as some generalized advantage function. Lemma 3.1 shows that the gradient of the objective relates to *differential Q-functions*. This inspires us to update the policy in the following mirror-descent style:

$$\pi_{k+1}(s) = \arg\min_{a\in\mathcal{A}} \bar{Q}^{\pi_k}(s,a) + h^a(s) + \frac{1}{\eta_k}D(\pi_k(s),a), \qquad (12)$$

where $\eta_k$ is some predefined step size. The complexity analysis of this type of algorithm has been thoroughly studied in [20] in tabular forms or linear function approximation settings, i.e., a lookup

table represents the policy, and the *differential Q-functions* are approximated as linear functions with respect to some feature space. In practice, especially in the Deep Reinforcement Learning community, both policy and the $Q$-functions are represented by neural networks. Thus, novel analysis is required for general function approximation. Additionally, the exact value of $\bar{Q}^\pi$ and $\nabla\omega(\pi)$ in Eq. 19 can only be estimated through approximation. Note that $\nabla\omega(\pi)$ rises inside the Bregman distance $D(\pi_k(s), a)$, referring to Eq. 13. We consider their stochastic estimators calculated from state-action sample pairs $\zeta$, denoted as $\bar{\mathcal{Q}}^{\pi,\zeta}(s, a; \phi)$, $\tilde{\nabla}\omega(\pi(s); \phi)$ respectively, where $\phi$ denotes the parameters of the method of choice, for example, weights and biases in a neural network. For the rest of the paper, we abbreviate $\bar{\mathcal{Q}}^{\pi,\zeta}(s, a; \phi)$ as $\bar{\mathcal{Q}}(s, a; \phi)$ when the context is clear. We described the Stochastic Policy Mirror Descent (SPMD) algorithm in Algorithm 1.

---

**Algorithm 1:** The Stochastic Policy Mirror Descent (SPMD) algorithm for AMDPs

---

1: **Input**: Initialize random policy $\pi_0$ and step size sequence $\{\eta_k\}$
2: **for** $k = 0, 1, \cdots, K$ **do**
3:     Sample collection: $\zeta_k = \{(s_t, a_t, c_t)\}_{t \geq 1}$
4:     **Critic step**: Approximate $\bar{Q}^{\pi_k}$, $\nabla\omega(\pi_k(s))$ with

$$\bar{Q}^{\pi_k}(s, a) \approx \bar{\mathcal{Q}}^{\pi_k, \zeta_k}(s, a; \phi_k), \tag{13}$$

$$\nabla\omega(\pi_k(s)) \approx \tilde{\nabla}\omega(\pi_k(s); \phi_k). \tag{14}$$

5:     **Actor step**: Update the policy

$$\pi_{k+1}(s) = \arg\min_{a \in \mathcal{A}} \bar{\mathcal{Q}}^{\pi_k, \zeta_k}(s, a; \phi_k) + h^a(s) + \tfrac{1}{\eta_k}(\langle\tilde{\nabla}\omega(\pi_k(s); \phi_k), a\rangle + \omega(a)). \tag{15}$$

6: **end for**

---

The following paragraphs present the complexity analysis for Algorithm 1. Note that we assume the approximated $Q$ function might not be convex but weakly convex, i.e.,

$$\bar{\mathcal{Q}}(s, a; \phi) + \mu_{\bar{\mathcal{Q}}}D(\pi_0(s), a) \tag{16}$$

is convex w.r.t. $a \in \mathcal{A}$ for some $\mu_{\bar{\mathcal{Q}}} \geq 0$ and $\pi_0$ is the initialized policy. This assumption is general as any differential function with Lipschitz continuous gradients is weakly convex. Moreover, we assume that $h^\cdot(s)$, $\bar{\mathcal{Q}}(s, \cdot; \theta)$ and $\bar{Q}^\pi(s, \cdot)$ are Lipschitz continuous with respect to the action.

**Assumption 3.2.** For all $a_1, a_2 \in \mathcal{A}$ there exist some constants $M_h, M_{\bar{\mathcal{Q}}}, M_{\bar{Q}}$ such that

$$|h^{a_1}(s) - h^{a_2}(s)| \leq M_h\|a_1 - a_2\|, \tag{17}$$

$$|\bar{\mathcal{Q}}(s, a_1; \phi) - \bar{\mathcal{Q}}(s, a_2; \phi)| \leq M_{\bar{\mathcal{Q}}}\|a_1 - a_2\|, \tag{18}$$

$$|\bar{Q}^\pi(s, a_1) - \bar{Q}^\pi(s, a_2)| \leq M_{\bar{Q}}\|a_1 - a_2\|, \tag{19}$$

and $\bar{\mathcal{Q}}(s, a; \phi)$ is $\mu_{\bar{\mathcal{Q}}}$-weakly convex, i.e., Eq. 16 is convex.

Note that the strong convexity modulus of the objective function in Eq. 15 can be very large since $\eta_k$ can be small, in which case the subproblem Eq. 15 is strongly convex, thus the solution of Eq. 15 is unique due to strong convexity [18].

Our complexity analysis follows the style of convex optimization. We begin the analysis by decomposing the approximation error in the following way

$$\delta_k^Q(s, a) := \bar{\mathcal{Q}}(s, a; \phi_k) - \bar{Q}^{\pi_k}(s, a), \tag{20}$$

$$\delta_k^\omega(s) := \tfrac{1}{\eta_k}\tilde{\nabla}\omega(\pi_k(s); \phi_k) - \tfrac{1}{\eta_k}\nabla\omega(\pi_k(s)),$$

$$= \mathbb{E}\left[\tfrac{1}{\eta_k}\tilde{\nabla}\omega(\pi_k(s); \phi_k)\right] - \tfrac{1}{\eta_k}\nabla\omega(\pi_k(s)) + \tfrac{1}{\eta_k}\tilde{\nabla}\omega(\pi_k(s); \phi_k) - \mathbb{E}\left[\tfrac{1}{\eta_k}\tilde{\nabla}\omega(\pi_k(s); \phi_k)\right]. \tag{21}$$

We refer to the first two and last two approximation error terms in $\delta^\omega$ as $\delta_k^{\omega, det}$ and $\delta_k^{\omega, sto}$, respectively, as they represent a deterministic approximation error which we cannot reduce and a stochastic error related to the variance of the estimator. They represent the noise introduced in the critic step. To make

the noise tractable and the entire algorithm convergent, we must assume they are not arbitrarily large. But before making more assumptions about these error terms, we must distinguish the case when $\bar{\mathcal{Q}} + h$ is convex and when it's not, i.e., when $\mu \geq 0$ or $\mu < 0$, where $\mu = \mu_h - \mu_{\bar{\mathcal{Q}}}$ for simplicity. We can obtain globally optimal solutions if $\mu \geq 0$. While in the other case, only stationary points can be obtained, e.g., both $\psi^{\pi}(s, \pi'(s))$ and $D(\pi_k, \pi_{k+1})$ approach 0. Note that $\psi$ relates to the progress we make in each iteration. See A.6 for more detail. These two cases require different assumptions on the error terms, so they must be treated differently. We start with the optimistic case when $\mu \geq 0$, under which we make the following assumption about the approximation error $\delta_k^{\bar{Q}}(s, a)$ and $\delta_k^{\omega}(s)$:

**Assumption 3.3.** When $\mu \geq 0$, the critic step has bounded errors, i.e., there exist some constants $\varsigma^{\bar{Q}}, \varsigma^{\omega}, \sigma^{\omega}$ such that

$$\mathbb{E}_{\zeta_k, s \sim \kappa^*}\left[|\delta_k(s, \pi_k(s))| + |\delta_k(s, \pi^*(s))|\right] \leq \varsigma^{\bar{Q}}, \tag{22}$$

$$\mathbb{E}_{\zeta_k, s \sim \kappa^*}[\|\delta_k^{\omega, det}(s)\|_*] \leq \varsigma^{\omega}, \mathbb{E}_{\zeta_k, s \sim \kappa^*}[\|\delta_k^{\omega, sto}\|_*^2] \leq (\sigma^{\omega})^2, \tag{23}$$

where $\zeta_k$ is the samples we collect in each iteration.

With the above assumption, we present the complexity result of SPMD when $\mu \geq 0$.

**Theorem 3.4.** *Suppose 2.1, 3.2, 3.3 hold and the step size $\eta_k$ satisfies*

$$\frac{\beta_k}{\eta_k} \leq \beta_{k-1}(\mu + \frac{1}{\eta_{k-1}}), k \geq 1, \tag{24}$$

*for some $\beta_k \geq 0$. Then after running the SPMD algorithm for $K$ iterations, we have*

$$\left(\sum_{k=0}^{K-1} \beta_k\right)^{-1} \left(\frac{1}{1-\Gamma}\sum_{k=0}^{K-1}\beta_k(\rho^{\pi_k} - \rho^*) + \beta_{K-1}(\mu + \frac{1}{\eta_{K-1}})\mathbb{E}[D(\pi_K, \pi^*)]\right)$$

$$\leq \left(\sum_{k=0}^{K-1}\beta_k\right)^{-1}\left(\frac{\beta_0}{\eta_0}D(\pi_0, \pi^*) + \sum_{k=0}^{K-1}\beta_k\eta_k[(2M_{\bar{\mathcal{Q}}} + M_{\bar{Q}} + M_h)^2/2 + (\sigma^{\omega})^2]\right)$$

$$+ \varsigma^{\bar{Q}} + \varsigma^{\omega}\bar{D}_{\mathcal{A}}, \tag{25}$$

*where $\bar{D}_{\mathcal{A}} := \max_{a_1, a_2 \in \mathcal{A}} D(a_1, a_2)$.*

Proof can be found in A.4. Equation 25 encapsulates a convergence overview without specifying explicit step size choices. To be specific, $\alpha_k, \beta_k$ are both step sizes, and the central term $\sum \rho_k - \rho^*$ (distance to the optimal value) plus $D(\pi_k, \pi^*)$ (the distance of the current policy to the optimal policy) will shrink as $k$ grows. The sum is bounded by some combination of $\alpha_k$ and $\beta_k$ plus an irreducible function approximation error $\varsigma$. The left-hand side is a weighted average of function value errors plus an average error from step 1 to $K$. The following result summarizes the convergence rate using particular step size choices. We obtain the first part of Corollary 3.5 by fixing the number of iteration $K$ and optimizing the right-hand side of Eq. 25. We get the rest by straightforward computation.

**Corollary 3.5.** *a) If $\mu = 0$, and $\eta_k = \sqrt{\frac{\mathcal{D}(\pi_1, \pi^*)}{K\left[\left(2M_{\bar{\mathcal{Q}}} + M_{\bar{Q}} + M_h\right)^2 + (\sigma^{\omega})^2\right]}}$ and $\beta_k = 1$, $k = 1, 2, 3 \ldots, K$, then the average reward will converge to the global optimal $\rho^*$*

$$\frac{1}{K}\sum_{k=0}^{K-1}(\rho^{\pi_k} - \rho^*)/(1 - \Gamma) = \mathcal{O}(K^{-1/2}) + \varsigma^{\bar{Q}} + \varsigma^{\omega}\bar{D}_{\mathcal{A}}. \tag{26}$$

*b) If $\mu > 0$, and $\eta_k = \frac{1}{\mu k}$ and $\beta_k = 1$, then the average reward will converge to the global optimal*

$$\frac{1}{K}\sum_{k=0}^{K-1}(\rho^{\pi_k} - \rho^*)/(1 - \Gamma) + \mu\mathbb{E}[D(\pi_K, \pi^*)] = \tilde{\mathcal{O}}(K^{-1}) + \varsigma^{\bar{Q}} + \varsigma^{\omega}\bar{D}_{\mathcal{A}}. \tag{27}$$

*c) If $\mu > 0$, $\eta_k = \frac{2}{\mu k}$ and $\beta_k = k + 1$, then the average reward will converge to the global optimal*

$$\frac{2}{K(K+1)}\sum_{k=0}^{K-1}\frac{k+1}{1-\Gamma}(\rho^{\pi_k} - \rho^*) + \mu\mathbb{E}[D(\pi_K, \pi^*)] = \mathcal{O}(K^{-2}) + \mathcal{O}(K^{-1}) + \varsigma^{\bar{Q}} + \varsigma^{\omega}\bar{D}_{\mathcal{A}}. \tag{28}$$

**Remark** The above result shows that if we carefully choose the function approximation $\bar{\mathcal{Q}}$ and the regularizer $h$ so that $\mu > 0$, we need $\mathcal{O}(\varepsilon^{-1})$ number of SPMD iterations to obtain $\varepsilon$ precision solution, i.e., $\frac{1}{K}\sum_{k=0}^{K-1}\rho_k^{\pi} - \rho^* + \mathbb{E}[D(\pi_K, \pi^*)] \leq \varepsilon$. The complexity deteriorates to $\mathcal{O}(\varepsilon^{-2})$ if $\mu = 0$. Notice that $\varsigma^{\bar{Q}}$ and $\varsigma^{\omega}$ are irreducible solely by policy optimization, as they arise due to function approximation and stochastic estimation errors.

When $\mu < 0$, global optimal solutions are unobtainable. The best we can do is to find a stationary point, that is when $\psi$ and $D(\pi_k, \pi_{k+1})$ tend to 0. In this case, we only need to assume that the estimation error terms $\delta_k^{\omega, det}(s)$ and $\delta_k^{\omega, sto}(s)$ are bounded, i.e.,

**Assumption 3.6.** When $\mu < 0$, the critic step has bounded errors for any $s \in \mathcal{S}$, i.e., there exist some constants $\bar{\varsigma}^\omega, \bar{\sigma}^\omega$ such that

$$\|\delta_k^{\omega,det}(s)\|_* \leq \bar{\varsigma}^\omega \quad \text{and} \quad \|\delta_k^{\omega,sto}(s)\|_* \leq \bar{\sigma}^\omega. \tag{29}$$

**Theorem 3.7.** *Let the number of iterations $K$ be fixed. Suppose that 2.1, 3.2, and 3.6 hold. Also assume that $\mu < 0$ and that $\eta_k = \eta = \min\{|\mu|/2, 1/\sqrt{K}\}, k = 0, \ldots, K - 1$. Then for any $s \in \mathcal{S}$, there exist iteration indices $k(s)$ found by running $K$ iterations of the SPMD method such that both the generalized advantage function and the distance between the iterates will converge to $0$, i.e.,*

$$\psi^{\pi_{k(s)}}(s, \pi_{k(s)+1}(s)) = \mathcal{O}(K^{-1}) + \mathcal{O}(K^{-1/2}) + \Gamma\bar{\varsigma}^\omega\bar{D}_\mathcal{A}/(1 - \Gamma) \tag{30}$$

$$\frac{1}{2\eta}D(\pi_{k(s)}, \pi_{k(s)+1}(s)) + (\mu + \frac{1}{\eta})D(\pi_{k(s)+1}(s), \pi_{k(s)}(s))$$
$$= \mathcal{O}(K^{-1}) + \mathcal{O}(K^{-1/2}) + \bar{\varsigma}^\omega\bar{D}_\mathcal{A}/(1 - \Gamma) \tag{31}$$

**Remark** This is the most general case, as we don't assume any restrictions on the function approximation class. As expected, the complexity bound becomes worse compared to previous results. To reach a $\varepsilon$-precision stationary point, we require at least $\mathcal{O}(\varepsilon^{-2})$ iterations. Note that $\psi \approx 0$ indicates that $\bar{Q}^{\pi_k}(s, \pi_{k+1}(s))$ has virtually no difference than $\bar{V}^{\pi_k}(s)$. This has a similar implication of $D(\pi_{k+1}(s), \pi_k(s))$ approaching $0$ which implies that we are reaching a stationary point.

## 3.1 Practical algorithm for Entropy Regularized AMDP

The above SPMD algorithm computes $\pi(s)$ for every state. In each iteration, we need to solve a subproblem Eq. 15 for every state (or for any state encountered). In practice, we prefer performing each SPMD iteration in a mini-batch style, i.e., approximately solving Eq. 15 from some trajectories collected. If the regularizer $h$ and $\omega$ are the negative entropy, we can approximately solve the SPMD actor step Eq. 15 by

$$\pi_{k+1} = \arg\min_{\pi \in \Pi} KL\left(\pi(s) \,\middle\|\, \frac{1}{Z(\phi_k)}\exp(\tilde{\mathcal{Q}}^{\pi_k}(s, a; \phi_k))\right), \tag{32}$$

where $Z(\phi_k)$ is some normalization constant and $\tilde{\mathcal{Q}}^{\pi_k}(s, a; \phi_k) := -\frac{\eta_k}{1+\eta_k}\bar{\mathcal{Q}}^{\pi_k}(s, a; \phi_k) - \frac{1}{1+\eta_k}\log\pi_k(s)$. In practice, we apply the so-called reparameterization trick [13] to obtain an unbiased gradient estimator. Let the policy be parameterized by

$$\pi(a|s) = \pi(f_\xi(\epsilon; s)|s), \forall(s, a) \in \mathcal{S} \times \mathcal{A}, \tag{33}$$

where $\epsilon$ is an input noise sampled from a fixed distribution; $\xi$ represents the parameters of the policy. Then we can approximately perform the actor step by solving the following problem:

$$\min_\xi \mathbb{E}_{(s,a)\sim\zeta,\epsilon}\left[\log(\pi(f_\xi(\epsilon; s)|s)) - \tilde{\mathcal{Q}}^{\pi_k}(s, f_\xi(\epsilon; s))\right]. \tag{34}$$

For large action spaces, the reparameterization trick efficiently approximates the solution of Eq. 15. As for the critic step, we can approximate $\bar{Q}$ by minimizing the Temporal Difference (TD) error, i.e., solving the following problem:

$$\min_\phi \mathbb{E}_{(s,a,c,s',a')\sim\zeta}\left(c(s, a) + h^a(s) - \tilde{\rho}^\pi + \bar{\mathcal{Q}}^{\pi,\zeta}(s', a'; \phi) - \bar{\mathcal{Q}}^{\pi,\zeta}(s, a; \phi)\right)^2, \tag{35}$$

where $\tilde{\rho}^\pi$ is the estimated average cost, e.g., by taking $\tilde{\rho}^\pi = \frac{1}{|\zeta|}\sum c(s, a) + h^a(s)$.

## 4 Inverse Policy Mirror Descent for IRL

Equipped with an efficient solver for AMDPs presented in section 3, we can now solve the IRL problem in the form of (Dual IRL). We suppose $\theta$ parameterizes the reward function with arbitrary methods, e.g., neural networks. To solve the dual problem, we update the parameter $\theta$ by performing a gradient descent step $\theta_{k+1} = \theta_k - \alpha_k\nabla_\theta L(\theta)$ where $\alpha_k$ is some predefined step size. The gradient computation of the dual objective function is

$$\nabla L(\theta) = \mathbb{E}_{(s,a)\sim d^E}\left[\nabla_\theta c(s, a; \theta)\right] - \mathbb{E}_{(s,a)\sim d^\pi}\left[\nabla_\theta c(s, a; \theta)\right]. \tag{36}$$

---

**Algorithm 2:** The Inverse Policy Mirror Descent (IPMD) algorithm

---

1: **Input**: Initialize random policy $\pi_0$ and step size sequence $\{\alpha_k\}$

2: **for** $k = 0, 1, \cdots, K$ **do**

3:     Collect samples $\{(s_t, a_t)\}_{t \geq 1}$ and compute $c(s_t, a_t; \theta_k)$ based on current reward estimation.

4:     **Critic step**: Implement a policy evaluation algorithm to evaluation $\bar{Q}^{\pi_k}$,

$$\bar{Q}^{\pi_k}(s, a) \approx \bar{\mathcal{Q}}^{\pi, \zeta_k}(s, a). \tag{37}$$

5:     **Actor step**: Update the policy by solving the following for every $s \in \mathcal{S}$

$$\pi_{k+1}(s) = \underset{\pi \in \Pi}{\arg\min} KL\left(\pi(s) \,\middle\|\, \tfrac{1}{Z} \exp(\tilde{\mathcal{Q}}^{\pi_k}(s, a))\right). \tag{38}$$

6:     **Dual Update step**: Perform the stochastic update

$$\theta_{k+1} = \theta_k - \alpha_k g_k. \tag{39}$$

7: **end for**

---

Without accessing the transition kernel P, the true gradient is unavailable. So we use a stochastic approximation instead, denoted as $g_k := g(\theta_k; \zeta_k^E) - g(\theta_k; \zeta_k^\pi)$ where $g(\theta; \zeta) := \frac{1}{N}\sum_{t=1}^N \nabla c(s_t, a_t; \theta)$ is the stochastic estimator of the gradient of the average reward. We describe the proposed Inverse Policy Mirror Descent (IPMD) algorithm in Algorithm 2. In this section, we provide our analysis that captures the algorithm behaviors across iterations since, in each iteration, $\bar{Q}$ is only an approximation of the true differential $Q$-function, which itself alters due to the change of the reward estimation. The idea of the proof is based on the Lipschitz continuity of the iterates, as it controls the difference of these functions across iterations. We make the following formal assumptions for such a purpose.

**Assumption 4.1.** For any $s \in \mathcal{S}, a \in \mathcal{A}$, the gradient of the reward function is bounded and Lipschitz continuous, i.e., there exist some constant real numbers $L_r, L_g$ so that the following holds:

$$\|\nabla c(s, a; \theta)\|_2 \leq L_r, \|\nabla c(s, a; \theta_1) - \nabla c(s, a; \theta_2)\|_2 \leq L_g\|\theta_1 - \theta_2\|_2.$$

Note that $\bar{Q}$ is also a function of $\theta$. As shown in section 3, $\bar{\mathcal{Q}}$ can be parameterized but we only denote the estimator as $\bar{\mathcal{Q}}^\pi(s, a; \theta_k)$ since its parameters do not contribute to the analysis.

**Assumption 4.2.** Suppose that at least one of $\mathcal{S}, \mathcal{A}$ is continuous, we assume that

$$\max_\theta \|\nabla_\theta \bar{\mathcal{Q}}^\pi(s, a; \theta)\|_2 \leq L_q, \tag{40}$$

where $L_q$ is some positive constant. For convenience we denote $\bar{\mathcal{Q}}^\pi(s, a; \theta_k)$ also as $\bar{\mathcal{Q}}^\pi_{\theta_k}(s, a)$.

We further assume that the estimation error from using $\bar{\mathcal{Q}}^\pi_\theta$ is bounded.

**Assumption 4.3.** For any $s \in \mathcal{S}, a \in \mathcal{A}$, there exist some constants $\varsigma_k, \varsigma, \nu_k, \nu$ such that the following inequalities hold:

$$\left\|\mathbb{E}_{\zeta_k}[\bar{\mathcal{Q}}^{\pi_k, \zeta_k}_{\theta_k}] - \bar{Q}^{\pi_k}_{\theta_k}\right\|_{sp,\infty} \leq \varsigma_k \leq \varsigma, \quad \mathbb{E}_{\zeta_k}\left[\left\|\bar{\mathcal{Q}}^{\pi_k, \zeta_k}_{\theta_k} - \bar{Q}^{\pi_k}_{\theta_k}\right\|^2_{sp,\infty}\right] \leq \nu_k^2 \leq \nu^2. \tag{41}$$

Notice that this error bound subsumes both the estimation errors and approximation errors. In general, the Bellman operator is nonexpansive in the average-reward setting, so analysis based on the contraction property of the Bellman operator with the infinity norm, as what is used in [37], fails in this case. To this end, we assume that the operator is span contractive, i.e.,

**Assumption 4.4.** In the critic step, there's a way to construct a span contractive Bellman operator $\mathcal{T}$ such that there exists $0 < \gamma < 1$ for any *differential Q-functions* $\bar{Q}_1, \bar{Q}_2$,

$$\|\mathcal{T}\bar{Q}_1 - \mathcal{T}\bar{Q}_2\|_{sp,\infty} \leq \gamma\|\bar{Q}_1 - \bar{Q}_2\|_{sp,\infty}. \tag{42}$$

In A.6, we provide conditions when the Bellman operator has a J-step span contraction. Now we can discuss the convergence results in the following theorems. The convergence of the *differential Q-function* is characterized in the following result.

**Theorem 4.5.** *Suppose that assumptions 2.1, 4.1-4.4 hold. If $\alpha = \frac{\alpha_0}{\sqrt{K}}$, the differential Q-function will converge to the optimal solution for a given reward function parameterized by $\theta_k$, i.e.,*

$$\frac{1}{K}\sum_{k=0}^{K-1}\left\|\mathbb{E}_{\zeta_k}[\bar{\mathcal{Q}}_{\theta_k}^{\pi_k,\zeta_k}] - \bar{Q}_{\theta_k}^{\pi_{\theta_k}}\right\|_{sp,\infty} = \mathcal{O}(K^{-1}) + \mathcal{O}(K^{-1/2}) + \frac{\varsigma}{1-\gamma}, \tag{43}$$

*where $\alpha_0 > 0$ is some step size chosen, and $\gamma$ is some constant defined in Assumption 4.4.*

Proof can be found in A.7. The above theorem shows that the *differential Q-function* will approach the optimal *differential Q-function* $\bar{Q}_{\theta_k}^{\pi_{\theta_k}}$ with respect to the current reward estimation $\theta_k$. If the reward estimation is accurate, so is the *differential Q-function*. Finally, in the next theorem, we show that the policy converges to the optimal policy for a given reward function, and the reward function approximation converges to a stationary point. Proof can be found in A.8.

**Theorem 4.6.** *Suppose that assumptions 2.1, 4.1-4.4 hold. If $\alpha = \frac{\alpha_0}{\sqrt{K}}$ and run the proposed IPMD algorithm $K$ iterations, the algorithm will produce near stationary solutions for the reward function and the optimal policy for such reward function. Specifically, the following holds,*

$$\frac{1}{K}\sum_{k=0}^{K-1}\|\mathbb{E}_{\zeta_k}[\log \pi_{k+1}] - \log \pi_{\theta_k}\|_\infty = \mathcal{O}(K^{-1}) + \mathcal{O}(K^{-1/2}) + \frac{\varsigma}{1-\gamma}, \tag{44}$$

$$\frac{1}{K}\mathbb{E}[\|\nabla L(\theta_k)\|_2^2] = \mathcal{O}(K^{-1}) + \mathcal{O}(K^{-1/2}) + \frac{1}{1-\gamma}2\nu L_c L_r C_d\sqrt{|\mathcal{S}|\cdot|\mathcal{A}|}, \tag{45}$$

*where $\alpha_0, \nu, L_c, L_r, C_d$ are some constants and $|\mathcal{S}|, |\mathcal{A}|$ some measure of the state and action space.*

**Remark** First, the above theorem shows that the policy will converge to the optimal policy given the reward parameterization, although each iteration's reward parameter differs. Second, the reward parameter $\theta$ will converge to a stationary point of (Dual IRL), as the objective can be highly nonconvex under general function approximation. Third, for general AMDPs without 1-step span contraction, we can use a $J$-step Bellman operator (see Appendix A.7) for policy evaluation to maintain a $\mathcal{O}(\varepsilon^{-2})$ complexity for the entire algorithm.

## 5 Numerical experiments

In this section, we showcase the performance of the proposed SPMD and IPMD algorithms. Our code can be found at `https://anonymous.4open.science/r/IPMD-9D60`. See more detail about all our implementation in the Appendix A.9.

**MuJoCo Robotics Manipulation Tasks for RL** This experiment tests our RL agent's performance on robotics manipulation tasks. Our SPMD algorithm is based on the `stable-baselines3` [28]. We compare the performance of our algorithm with Soft Acrot-Critic (SAC) [13] implemented in [28]. The policy network employs two fully connected hidden layers of dimension 256 each, taking actions as input and outputting a distribution. Both the $Q$ network and reward function share the same architecture, with ReLU activation used in hidden layers. A double $Q$-learning technique is used to minimize overestimation [13]. During training, we found that setting the entropy coefficient term to 0.01 makes training stable and efficient. The learning rate is $3e^{-4}$. Each step of the algorithm samples 512 state-action sample pairs. Table 1 reports the numerical results of each model. Our proposed SPMD achieves on-par performance with SAC and exceptionally better performance in the Humanoid environment.

**MuJoCo benchmark for IRL** In this experiment, we compare the proposed IPMD method with IQ-Learn [11] and $f$-IRL [23]. The authors of ML-IRL have not released its implementation at the time we experimented. Nevertheless, the performance of ML-IRL is comparable to $f$-IRL. It is observed in the literature that imitation learning algorithms have inferior performance [37]. Hence we omit these methods. Table 2 reports the numerical results of each model. Note that we cannot record a competitive result for IQ-Learn with Humanoid as claimed in the original paper, which we highlight using $*$. The result shows that IPMD outperforms in a variety of environments. One possible reason Ant is falling behind is that Ant has more ground contact since it has more legs. This will impact the mixing time of the MDP and our assumption on the 1-step contraction. Compared to Half-Cheetah, or Humanoid, it is harder for the Ant to transition from an arbitrary state to another arbitrary state as it involves multiple legs working together. One remedy is that we construct a $J$-step contractive operator for policy evaluation, as done in [7]. The success of humanoid is possibly due to a slightly different policy evaluation scheme compared to the discounted setting, where the entropy

| Table 1. MuJoCo Results for RL. The average performance of RL agents across five runs. | | |
| --- | --- | --- |
| Task | SAC | SPMD (Ours) |
| Hopper | $3876 \pm 78$ | $3619 \pm 5$ |
| Half-Cheetah | $13300 \pm 46$ | $13025 \pm 37$ |
| Walker | $6115 \pm 41$ | $4454 \pm 26$ |
| Ant | $5118 \pm 57$ | $5230 \pm 118$ |
| Humanoid | $6923 \pm 11$ | $10249 \pm 7$ |

| Table 2. MuJoCo Results for IRL. The average performance of IRL agents across five runs. | | | |
| --- | --- | --- | --- |
| IQL | $f$-IRL | IPMD (ours) | Expert |
| 1909 | 3083 | **3564** | 4115 |
| 9132 | 11259 | **12634** | 15467 |
| 5155 | 5378 | **5423** | 5323 |
| 3486 | **5460** | 4053 | 5783 |
| 661* | 3580 | **7379** | 7137 |

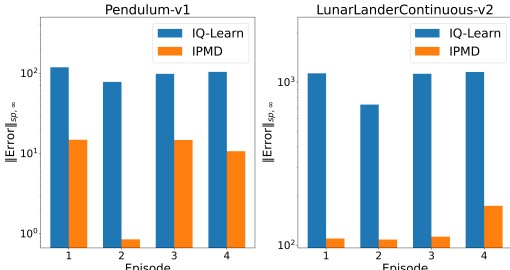

**Figure 1.** Reward recovery performance of IPMD and IQ-Learn. The lower the bar, the better the performance.

term from the policy no longer plays a part, as the term $c - \rho$ cancels out the additional regularization and entropy of the policy. We suspect this brings more stable training and thus higher performance.

**Reward Recovery** Finally, we compare the proposed IPMD method against IQ-Learn on recovering the expert's reward function in two environments: Pendulum and LunarLanderContinuous, both implemented in [4]. The result of the experiment is shown in Figure 1. Using state-action pairs from 4 different episodes, we compare the span seminorm of the predicted reward and the true reward (the reward the expert uses). The experts are trained with a discount factor $0.99$. IQ-Learn's discount factor is set to $0.99$. The result shows IPMD's superiority in reward recovery.

## 6 Discussion

In this paper, we formulate the Maximum Entropy IRL problem with the average-reward criterion and propose efficient algorithms to solve it. We devise the SPMD method for solving RL problems in general state and action spaces beyond the scope of linear function approximation with strong theoretical guarantees. We integrate this method to develop the IPMD method for solving the entire IRL problem and provide convergence analysis. To reach a $\varepsilon$ precision stationary point, IPMD requires $\mathcal{O}(\varepsilon^{-2})$ policy optimization steps.

We also notice there are possible improvements from this work. For example, we impose various assumptions both the function approximation classes and strong reliance on the accuracy of the critic step. It is natural to seek more robust methods when facing inaccurate estimations. Additionally, we impose continuity assumption in various Assumptions (e.g. Assump 4.2). Such assumptions can be violated if neural networks are used as function approximations. In practice, it is possible to alleviate the issue by "clipping" the gradient norm when performing gradient descent, which proves effective in related works [30]. Furthermore, we only consider uniform ergodic chains. It is interesting to see how our method would hold under the unichain or multi-chain setting. Lastly, the analysis based on span contraction might be improved for general MDPs. This requires nontrivial work and thus we leave it for future development.

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
