# A Appendix A

## A.1 Proof of Lemma 3.1

Assumption 2.1 implies that for any policy $\pi$, $\rho(\pi) = \mathbb{E}[c(s,a) + h^\pi(s) \mid a \sim \pi(\cdot|s), s \sim \nu^\pi]$.

$$
\begin{aligned}
\rho^{\pi'} - \rho^\pi &= \lim_{T\to\infty} \tfrac{1}{T}\mathbb{E}\Big[\textstyle\sum_{t=0}^{T-1} r(s_t,a_t) + h^{\pi'(s_t)}(s_t) - \rho(\pi) + \bar{V}^\pi(s_{t+1}) - \bar{V}^\pi(s_t) \\
&\qquad + h^{\pi(s_t)}(s_t) - h^{\pi(s_t)}(s_t) + \rho(\pi) \mid s_0 = s, a_t \sim \pi'(\cdot|s_t), s_{t+1} \sim \mathsf{P}(\cdot|s_t,a_t)\Big] \\
&\qquad + \lim_{T\to\infty} \tfrac{1}{T}\mathbb{E}[\bar{V}^\pi(s_0) \mid s_0 = s] - \rho(\pi) \\
&= \lim_{T\to\infty} \tfrac{1}{T}\mathbb{E}\Big[\textstyle\sum_{t=0}^{T-1} \bar{Q}^\pi(s_t,a_t) - \bar{V}^\pi(s_t) \\
&\qquad + h^{\pi'(s_t)}(s_t) - h^{\pi(s_t)}(s_t) \mid s_0 = s, a_t \sim \pi'(\cdot|s_t), s_{t+1} \sim \mathsf{P}(\cdot|s_t,a_t)\Big] \\
&= \int \bar{Q}^\pi(s',\pi'(s')) - \bar{V}^\pi(s') + h^{\pi'(s)}(s') - h^{\pi(s')}(s')\kappa^{\pi'}(ds')
\end{aligned}
$$

## A.2 Optimality condition

The following lemma uses the optimality condition of (12), serving an important recursion for the deterministic case. We denote convexity modulus of $\bar{Q}$ by $\mu_Q$ and define $\mu_d := \mu_h - \mu_Q$.

**Lemma A.1.** *[19, Lemma 3] If $\eta_k$ in (15) satisfies*

$$
\mu_d + \tfrac{1}{\eta_k} \geq 0, \tag{46}
$$

*then for any $a \in \mathcal{A}$,*

$$
\begin{aligned}
\psi^{\pi_k}(s,\pi_{k+1}(s)) &+ \tfrac{1}{\eta_k}D(\pi_k(s),\pi_{k+1}(s)) \\
&+ (\mu_d + \tfrac{1}{\eta_k})D(\pi_{k+1}(s),a) \\
&\leq \psi^{\pi_k}(s,a) + \tfrac{1}{\eta_k}D(\pi_k(s),a). \tag{47}
\end{aligned}
$$

## A.3 Progress in each iteration

**Proposition A.2.** *For any $s \in \mathcal{S}$, we have*

$$
\begin{aligned}
\tfrac{1}{1-\Gamma}\left(\rho(\pi_{k+1}) - \rho(\pi_k)\right) &\leq \psi^{\pi_k}(s,\pi_{k+1}(s)) \\
&\leq -[\tfrac{1}{\eta_k}D(\pi_k(s),\pi_{k+1}(s)) + (\mu_d + \tfrac{1}{\eta_k})D(\pi_{k+1}(s),\pi_k(s))]. \tag{48}
\end{aligned}
$$

The above proposition shows that the progress of each step relates to the advantage function.

*Proof.* By the above lemma with $a = \pi_k(s)$, we have

$$
\begin{aligned}
\psi^{\pi_k}(s,\pi_{k+1}(s)) &+ \tfrac{1}{\eta_k}D(\pi_k(s),\pi_{k+1}(s)) + (\mu_d + \tfrac{1}{\eta_k})D(\pi_{k+1}(s),\pi_k(s)) \\
&\leq \psi^{\pi_k}(s,\pi_k(s)) + \tfrac{1}{\eta_k}D(\pi_k(s),\pi_k(s)) = 0, \tag{49}
\end{aligned}
$$

where the last identity follows from the fact that $\psi^{\pi_k}(s,\pi_k(s)) = 0$ due to Assumption (6) and (10). By Lemma 3.1, Eq. 49 and the fact that $\kappa_s^{\pi_{k+1}}(\{s\}) \geq 1 - \Gamma$ due to (2.1), we have

$$
\begin{aligned}
\rho(\pi_{k+1}) - \rho(\pi_k) &= \int \psi^{\pi_k}(q,\pi_{k+1}(q))\kappa_s^{\pi_{k+1}}(dq) \\
&\leq \psi^{\pi_k}(s,\pi_{k+1}(s))\,\kappa_s^{\pi_{k+1}}(\{s\}) \\
&\leq (1-\Gamma)\psi^{\pi_k}(s,\pi_{k+1}(s)). \tag{50}
\end{aligned}
$$

The result then follows by combining the above two inequalities. $\qquad\square$

## A.4 Proof of Theorem 3.4

*Proof.* Let $\mathfrak{L}^{\pi_k}(s,a) := \bar{Q}^{\pi_k}(s,a) + \frac{1}{\eta_k}\langle\nabla\omega(\pi_k(s)),a\rangle$ and $\mathcal{L}(s,a;\theta_k)$ be its stochastic estimator. Denote $\delta(s,a) := \psi_k(s,a) - \mathfrak{L}_k(s,a)$. By the optimality condition of (15), we have

$$\mathcal{L}(s,\pi_{k+1}(s);\theta_k) - \mathcal{L}(s,a;\theta_k) + h^{\pi_{k+1}(s)}(s) - h^a(s)$$
$$+ \frac{1}{\eta_k}[\omega(\pi_{k+1}(s)) - \omega(a)] + (\mu + \frac{1}{\eta_k})D(\pi_{k+1}(s),a) \leq 0, \quad (51)$$

which implies that

$$\mathfrak{L}_k(s,\pi_{k+1}(s)) - \mathfrak{L}_k(s,a) + [h^{\pi_{k+1}(s)}(s) - h^a(s)] + \frac{1}{\eta_k}[\omega(\pi_{k+1}(s)) - \omega(a)]$$
$$+ (\mu + \frac{1}{\eta_k})D(\pi_{k+1}(s),a) + \delta_k^Q(s,\pi_{k+1}(s)) - \delta_k^Q(s,a) + \langle\delta_k^\omega(s),\pi_{k+1}(s) - a\rangle \leq 0. \quad (52)$$

In view of the definition (10), we can show that

$$\psi^{\pi_k}(s,\pi_{k+1}(s)) - \psi^{\pi_k}(s,a) + h^{\pi_{k+1}(s)}(s) - h^{\pi_k(s)}(s) + \frac{1}{\eta_k}[D(\pi_k(s),\pi_{k+1}(s)) - D(\pi_k(s),a)]$$
$$+ (\tilde{\mu}_d + \frac{1}{\eta_k})D(\pi_{k+1}(s),a) + \delta_k^Q(s,\pi_{k+1}(s)) - \delta_k^Q(s,a) + \langle\delta_k^\omega(s),\pi_{k+1}(s) - a\rangle \leq 0, \quad (53)$$

using the following inequalities and the definition of Bregman distance

$$\bar{Q}^{\pi_k}(s,\pi_{k+1}(s)) - \bar{Q}^{\pi_k}(s,\pi_k(s)) \geq -M_{\bar{Q}}\|\pi_{k+1}(s) - \pi_k(s)\|, \quad (54)$$

$$h^{\pi_{k+1}(s)}(s) - h^{\pi_k(s)}(s) \geq -M_h\|\pi_{k+1}(s) - \pi_k(s)\|, \quad (55)$$

$$\delta_k^Q(s,\pi_{k+1}(s)) = \delta_k^Q(s,\pi_{k+1}(s)) - \delta_k^Q(s,\pi_k(s)) + \delta_k^Q(s,\pi_k(s))$$
$$\geq -(M_{\tilde{Q}} + M_{\bar{Q}}\|\pi_{k+1}(s) - \pi_k(s)\| + \delta_k^Q(s,\pi_k(s)), \quad (56)$$

$$\langle\delta_k^\omega(s),\pi_{k+1}(s) - a\rangle = \langle\delta_k^{\omega,det}(s),\pi_{k+1}(s) - a\rangle + \langle\delta_k^{\omega,sto}(s),\pi_k(s) - a\rangle$$
$$+ \langle\delta_k^{\omega,sto}(s),\pi_{k+1}(s) - \pi_k(s)\rangle$$
$$\geq -\|\delta_k^{\omega,det}(s)\|_*\bar{D}_{\mathcal{A}} - \|\delta_k^{\omega,sto}(s)\|_*\|\pi_{k+1}(s) - \pi_k(s)\|$$
$$+ \langle\delta_k^{\omega,sto}(s),\pi_k(s) - a\rangle, \quad (57)$$

where the first three inequalities are from the definition of Lipschitz continuity, and the last inequality is from Cauchy-Schwartz inequality. Notice the approximation errors are all captured by the error terms. We then conclude from the above inequality and Young's inequality that

$$-\psi^{\pi_k}(s,a) + (\mu + \frac{1}{\eta_k})D(\pi_{k+1}(s),a)$$
$$\leq \frac{1}{\eta_k}D(\pi_k(s),a) + \frac{\eta_k}{2}(2M_Q + M_{\tilde{Q}} + M_h + \|\delta_k^{\omega,sto}(s)\|_*)^2$$
$$- \delta_k^Q(s,\pi_k(s)) + \delta_k^Q(s,a) + \|\delta_k^{\omega,det}(s)\|_*\bar{D}_{\mathcal{A}}$$
$$- \langle\delta_k^{\omega,sto}(s),\pi_k(s) - a\rangle. \quad (58)$$

Note that setting $a = \pi^*$ and taking conditional expectation of the above inequality w.r.t. $\xi_k$, it then follows from Lemma A.2, Equation 23 and $\mathbb{E}_{\xi_k}[\delta_k^{sto}(s)] = 0$ that

$$\frac{1}{1-\Gamma}(\rho(\pi_k) - \rho^*) + (\mu + \frac{1}{\eta_k})\mathbb{E}_{\xi_k}[\mathbb{E}_{s\sim\nu^*}D(\pi_{k+1}(s),\pi^*(s))]$$
$$\leq \frac{1}{\eta_k}\mathbb{E}_{s\sim\nu^*}D(\pi_k(s),\pi^*(s)) + \frac{\eta_k}{2}[(M_{\tilde{Q}} + M_{\bar{Q}} + M_h)^2 + (\sigma^\omega)^2]$$
$$+ \mathbb{E}_{\xi_k}\{\mathbb{E}_{s\sim\nu^*}[|\delta_k(s,\pi_k(s))| + |\delta_k(s,\pi_*(s))|]\} + \|\delta_k^{\omega,det}(s)\|_*\bar{D}_{\mathcal{A}}. \quad (59)$$

Note that the term $D(\pi_k(s),\pi^*)$ follows a telescopic sequence with the step size choice (24). Taking full expectation w.r.t. $\xi_k$ and the $\beta_k$-weighted sum of the above inequalities, it then follows from (24) and the definition of $\rho$ that

$$\frac{1}{1-\Gamma}\sum_{k=0}^{K-1}\beta_t\mathbb{E}[(\rho(\pi_k) - \rho^*)] + \beta_{K-1}(\mu + \frac{1}{\eta_{K-1}})\mathbb{E}_{\xi_k,s\sim\nu^*}D(\pi_k(s),\pi^*(s))$$
$$\leq \frac{\beta_0}{\eta_0}\mathbb{E}_{s\sim\nu^*}D(\pi_0(s),\pi^*(s)) + \sum_{k=0}^{K-1}\beta_k\eta_k[(2M_{\tilde{Q}} + M_{\bar{Q}} + M_h)^2/2 + (\sigma^\omega)^2]$$
$$+ (\varsigma^{\bar{Q}} + \varsigma^\omega\bar{D}_{\mathcal{A}})(\sum_{k=0}^{K-1}\beta_k), \quad (60)$$

from which we get the desired result by dividing both sides $\sum_{k=0}^{K-1}\beta_t$. $\qquad\square$

## A.5 Proof of Theorem 3.7

*Proof.* Setting $a = \pi_k(s)$ in (53), and using the facts that $\psi^{\pi_k}(s, \pi_k(s)) = 0$ and $D(\pi_k(s), \pi_k(s)) = 0$, we obtain

$$\psi^{\pi_k}(s, \pi_{k+1}(s)) + h^{\pi_{k+1}(s)}(s) - h^{\pi_k}(s) + \frac{1}{\eta_k} D(\pi_k(s), \pi_{k+1}(s))$$
$$+ (\tilde{\mu}_d + \frac{1}{\eta_k}) D(\pi_{k+1}(s), \pi_k(s)) + \delta_k^Q(s, \pi_{k+1}(s))$$
$$- \delta_k^Q(s, a) + \langle \delta_k^\omega(s), \pi_{k+1}(s) - a \rangle \leq 0. \tag{61}$$

Using the above inequality, (55)-(57), and Young's inequality, we then have

$$\psi^{\pi_k}(s, \pi_{k+1}(s)) + \frac{1}{2\eta_k} D(\pi_k(s), \pi_{k+1}(s)) + (\tilde{\mu}_d + \frac{1}{\eta_k}) D(\pi_{k+1}(s), \pi_k(s))$$
$$\leq (M_h + M_{\tilde{Q}} + M_{\bar{Q}} + \|\delta_k^{\omega,sto}(s)\|_*) \|\pi_{k+1}(s) - \pi_k(s)\| - \frac{1}{2\eta_k} D(\pi_k(s), \pi_{k+1}(s))$$
$$+ \|\delta_k^{\omega,det}(s)\|_* \bar{D}_{\mathcal{A}}$$
$$\leq \eta_k (M_h + M_{\tilde{Q}} + M_{\bar{Q}} + \|\delta_k^{\omega,sto}(s)\|_*)^2 + \|\delta_k^{\omega,det}(s)\|_* \bar{D}_{\mathcal{A}}.$$
$$\leq \eta_k (M_h + M_{\tilde{Q}} + M_{\bar{Q}} + \bar{\sigma}^\omega)^2 + \bar{\varsigma}^\omega \bar{D}_{\mathcal{A}}. \tag{62}$$

Similar to Eq. 49, we can show that

$$\rho(\pi_{k+1}) - \rho(\pi_k) = \int \psi^{\pi_k}(q, \pi_{k+1}(q))] \kappa_s^{\pi_{k+1}}(dq)$$
$$= \int \psi^{\pi_k}(q, \pi_{k+1}(q)) - \left[ \eta_k (M_h + M_{\tilde{Q}} + M_{\bar{Q}} + \bar{\sigma}^\omega)^2 + \bar{\varsigma}^\omega \bar{D}_{\mathcal{A}} \right] \kappa_s^{\pi_{k+1}}(dq)$$
$$+ \eta_k (M_h + M_{\tilde{Q}} + M_{\bar{Q}} + \bar{\sigma}^\omega)^2 + \bar{\varsigma}^\omega \bar{D}_{\mathcal{A}}$$
$$\leq (1 - \Gamma)\psi^{\pi_k}(s, \pi_{k+1}(s)) + \Gamma \left[ \eta_k (M_h + M_{\tilde{Q}} + M_{\bar{Q}} + \bar{\sigma}^\omega)^2 + \bar{\varsigma}^\omega \bar{D}_{\mathcal{A}} \right] \tag{63}$$

The result in (30) follows by taking the telescopic sum of the above inequality, while the one in (31) follows directly from (30) and the first inequality we proved in this subsection (62). $\qquad\square$

## A.6 Proof of an Auxiliary Lemma

To start the analysis, we first need a lemma to bound the total variation norm of policies with respect to its parameterization. This lemma will be used later to bound the convergence of the differential Q-function. Notice that for ergodic chains. we can obtain the following lemma by replacing $\tilde{P}$ with $P$ in the original proof.

**Lemma A.3.** *[Lemma 3 [33]] Consider the initialization distribution $\eta(\cdot)$ and transition kernel $\mathcal{P}(\cdot|s,a)$. Under $\eta(\cdot)$ and $\mathcal{P}(\cdot|s,a)$, denote $d_w(\cdot,\cdot)$ as the state-action visitation distribution of the MDP with the Boltzmann policy parameterized by the parameter $w$, i.e., $\pi \propto \exp w$. Suppose Assumption 2.1 holds, for all policy parameter $w$ and $w'$, we have*

$$\|d_w(\cdot,\cdot) - d_{w'}(\cdot,\cdot)\|_{TV} \leq C_d \|w - w'\|_2 \tag{64}$$

*where $C_d$ is a positive constant.*

In the next step, we show that the gradient of the differential $Q$-function and the gradient of the dual objective is also bounded and Lipschitz continuous. For discrete state and action spaces, we adopt standard stochastic matrix theory. However, we need to make the following assumption for continuous state and action spaces.

**Lemma A.4.** *For any $s \in \mathcal{S}, a \in \mathcal{A}$ the differential Q-function and the gradient of the objective in Dual IRL is smooth, i.e.,*

$$|\bar{Q}^{\pi_{\theta_1}}(s,a) - \bar{Q}^{\pi_{\theta_2}}(s,a)| \leq L_q \|\theta_1 - \theta_2\|_2, \tag{65}$$

$$\|\nabla_\theta L(\theta_1) - \nabla_\theta L(\theta_2)\|_2 \leq L_c \|\theta_1 - \theta_2\|_2, \tag{66}$$

*for some constant real number $L_q$ and $L_c$.*

*Proof.* Note that from (7) and Appendix A from [27] we have for any given policy $\pi$,

$$\nabla_\theta \bar{Q} = \nabla_\theta c - \nabla_\theta \rho + P^\pi (I - P^\pi + P^*)^{-1}(I - P^*)\nabla_\theta c \tag{67}$$

where $P^*$ is the limiting matrix of the transition kernel $P^\pi$, i.e., $P = \lim_{N \to \infty} \frac{1}{N}\sum_{n=1}^{N} P^n$, and $P^n$ is the n-th power of the matrix $P$. Since the chain is uniformly ergodic (ref to ergodicity) we have $P^* = e\kappa^T$, i.e.,

$$P^* = \begin{bmatrix} - & \kappa^T & - \\ - & \kappa^T & - \\ & \vdots & \end{bmatrix}. \tag{68}$$

Taking the gradient of both sides, we have

$$\nabla_\theta \bar{Q} = \nabla_\theta c - \nabla_\theta \rho + P^\pi (I - P^\pi + P^*)^{-1}(I - P^*)\nabla_\theta c. \tag{69}$$

To bound the last term, we first note that $\|P^\pi\|_2 = \|I - P^*\|_2 = 1$. Furthermore, we have for ergodic Markov chains, $\lambda_{\min}(I - P + P^*) > 0$ and the following decomposition (Theorem A.5, [27])

$$I - P^\pi + P^* = W^{-1}\begin{bmatrix} I - Q & 0 \\ 0 & I \end{bmatrix} W \tag{70}$$

Denote $1/\beta := \lambda_{\min}(I - Q)$, we have for any $s \in \mathcal{S}, a \in \mathcal{A}$

$$\begin{aligned}
\|\nabla_\theta \bar{Q}\|_2 &= \left\|\nabla_\theta c - \nabla_\theta \rho + P^\pi (I - P^\pi + P^*)^{-1}(I - P^*)\nabla_\theta c\right\|_2 \\
&\leq \|\nabla_\theta c\|_2 + \|\nabla_\theta \rho\|_2 + \left\|P^\pi (I - P^\pi + P^*)^{-1}(I - P^*)\nabla_\theta c\right\|_2 \\
&\leq 2\|\nabla_\theta c\|_2 + \left\|P^\pi (I - P^\pi + P^*)^{-1}(I - P^*)\right\|_2 \|\nabla_\theta c\|_2 \\
&\leq 2\|\nabla_\theta c\|_2 + \|P^\pi\|_2 \|(I - P^\pi + P^*)^{-1}\|_2 \|(I - P^*)\|_2 \|\nabla_\theta c\|_2 \\
&\leq 2\|\nabla_\theta c\|_2 + \beta\|\nabla_\theta c\|_2 \\
&\leq (2 + \beta)L_r
\end{aligned} \tag{71}$$

where the last step follows Assumption 4.1. Using the mean value theorem, we get

$$\begin{aligned}
\left|\bar{Q}^{\pi_{\theta_1}}(s,a;\theta_1) - \bar{Q}^{\pi_{\theta_2}}(s,a;\theta_2)\right| &\leq \max_\theta \|\nabla_\theta \bar{Q}^{\pi_\theta}(s,a;\theta)\|\|\theta_1 - \theta_2\| \\
&\leq (2 + \beta)L_r\|\theta_1 - \theta_2\| 
\end{aligned} \tag{72}$$

Denoting $L_q = 2 + \beta$ and taking the minimum over all state-action pairs, we get the desired result for discrete state and action. For general state and action spaces, in view of Assumption 4.2, it is easy to verify the result. Note that from the above analysis, the boundedness of the gradient under general state and action spaces is not too restrictive.

For the second part, note that

$$\nabla_\theta L(\theta_1) - \nabla_\theta L(\theta_2) = \mathbb{E}_{(s,a) \sim d^E} \left[ \nabla_\theta c(s, a; \theta_1) \right] - \mathbb{E}_{(s,a) \sim d^{\pi_{\theta_1}}} \left[ \nabla_\theta c(s, a; \theta_1) \right]$$
$$+ \mathbb{E}_{(s,a) \sim d^E} \left[ \nabla_\theta c(s, a; \theta_2) \right] - \mathbb{E}_{(s,a) \sim d^{\pi_{\theta_2}}} \left[ \nabla_\theta c(s, a; \theta_2) \right] \tag{73}$$

Using triangle inequality, we have

$$\left\| \mathbb{E}_{(s,a) \sim d^E} \left[ \nabla_\theta c(s, a; \theta_1) - \nabla_\theta c(s, a; \theta_2) \right] \right\|_2 \leq \mathbb{E}_{(s,a) \sim d^E} \left\| \nabla_\theta c(s, a; \theta_1) - \nabla_\theta c(s, a; \theta_2) \right\|$$
$$= L_g \left\| \theta_1 - \theta_2 \right\|_2 \tag{74}$$

Additionally,

$$\left\| \mathbb{E}_{d_1} \left[ \nabla_\theta c(s, a; \theta_1) \right] - \mathbb{E}_{d_2} \left[ \nabla_\theta c(s, a; \theta_1) \right] \right\|_2$$
$$\leq \left\| \mathbb{E}_{d_1} \left[ \nabla_\theta c(s, a; \theta_1) - \nabla_\theta c(s, a; \theta_2) \right] \right\|_2 + \left\| \mathbb{E}_{d_1} \left[ \nabla_\theta c(s, a; \theta_2) \right] - \mathbb{E}_{d_2} \left[ \nabla_\theta c(s, a; \theta_2) \right] \right\|_2 \tag{75}$$
$$\overset{(i)}{\leq} \mathbb{E}_{d_1} \left\| \nabla_\theta c(s, a; \theta_1) - \nabla_\theta c(s, a; \theta_2) \right\|_2 + 2 \max(\left\| \nabla_\theta c \right\|_2) \cdot \left\| d^{\pi_{\theta_1}}(\cdot, \cdot) - d^{\pi_{\theta_2}}(\cdot, \cdot) \right\|_{TV}$$
$$\overset{(ii)}{\leq} \mathbb{E}_{d_1} L_g \left\| \theta_1 - \theta_2 \right\|_2 + 2 L_r C_d \left\| \bar{Q}_{\theta_1}^{\pi_{\theta_1}} - \bar{Q}_{\theta_2}^{\pi_{\theta_2}} \right\|_2$$
$$\overset{(iii)}{\leq} \mathbb{E}_{d_1} L_g \left\| \theta_1 - \theta_2 \right\|_2 + 2 L_r C_d \sqrt{|\mathcal{S}| \cdot |\mathcal{A}|} \left\| \bar{Q}_{\theta_1}^{\pi_{\theta_1}} - \bar{Q}_{\theta_2}^{\pi_{\theta_2}} \right\|_\infty$$
$$\leq \left( L_g + 2 L_q L_r C_d \sqrt{|\mathcal{S}| \cdot |\mathcal{A}|} \right) \left\| \theta_1 - \theta_2 \right\|_2 \tag{76}$$

where $(i)$ follows from the Hölder's inequality; $(ii)$ follows from Lemma A.3; $(iii)$ follows from the equivalence of $L^p$ norms, which is also from the Hölder's inequality. Combining the two inequalities,

$$\left\| \nabla_\theta L(\theta_1) - \nabla_\theta L(\theta_2) \right\|_2 \leq \left( 2 L_g + 2 L_q L_r C_d \sqrt{|\mathcal{S}| \cdot |\mathcal{A}|} \right) \left\| \theta_1 - \theta_2 \right\|_2 \tag{77}$$

Denoting $L_c := 2 L_g + 2 L_q L_r C_d \sqrt{|\mathcal{S}| \cdot |\mathcal{A}|}$ we have the claimed result. $\qquad\square$

We also need Lipschitz continuity of the action-value function w.r.t $\theta$. We can easily get the following lemma in view of Lemma A.4, combined with the mean value theorem.

**Lemma A.5.** *Under any feasible policy $\pi$, for any $s \in \mathcal{S}$, $a \in \mathcal{A}$ and reward function parameters $\theta_1, \theta_2$, the following holds*

$$\left| \bar{Q}^\pi(s, a; \theta_1) - \bar{Q}^\pi(s, a; \theta_2) \right| \leq L_q \left\| \theta_1 - \theta_2 \right\|_2 \tag{78}$$

*where $L_q$ is defined in A.4.*

## A.7 Proof of Theorem 4.5

To show the convergence of the action-value function, we first need to show that the policy evaluation step is span contractive, which has a similar proof of Theorem 6.6.6 in [27]. For convenience, we define a 1-step Bellman operator for a given policy $\pi$ as

$$\mathcal{T}\bar{Q}(s,a) := c(s,a) - \rho(\pi) + P_\pi(s',a'|s,a)\bar{Q}(s',a'). \tag{79}$$

**Lemma A.6.** *Define $\gamma$ by*

$$\gamma := 1 - \min_{(s_1,a_1),(s_2,a_2)} \sum_{s\in\mathcal{S}, a\in\mathcal{A}} \min\{P_\pi(s,a|s_1,a_1), P_\pi(s,a|s_2,a_2)\} \tag{80}$$

*Then for any two differential Q-functions $p, q$,*

$$\|\mathcal{T}p - \mathcal{T}q\|_{sp,\infty} \leq \gamma \|p - q\|_{sp,\infty} \tag{81}$$

*Furthermore, if for any state-action pairs $(s_1, a_1)$ and $(s_2, a_2)$ there exists a third state-action pair $(s', a')$ such that both $P_\pi(s', a'|s_1, a_1) > 0$ and $P_\pi(s', a'|s_2, a_2) > 0$, then $\gamma < 1$.*

*Proof.* Let $(s^*, a^*) := \arg\max_{(s,a)\in\mathcal{S}\times\mathcal{A}} \mathcal{T}p - \mathcal{T}q$ and $(s_*, a_*) := \arg\min_{(s,a)\in\mathcal{S}\times\mathcal{A}} \mathcal{T}p - \mathcal{T}q$ so that

$$(\mathcal{T}p)(s^*, a^*) - (\mathcal{T}q)(s^*, a^*) = P_\pi(p-q)(s^*, a^*)$$
$$(\mathcal{T}p)(s_*, a_*) - (\mathcal{T}q)(s_*, a_*) = P_\pi(p-q)(s_*, a_*)$$

from which we get

$$\begin{aligned}
\|\mathcal{T}p - \mathcal{T}q\|_{sp,\infty} &= \max_{(s,a)\in\mathcal{S}\times\mathcal{A}} \mathcal{T}(p-q) - \min_{(s,a)\in\mathcal{S}\times\mathcal{A}} \mathcal{T}(p-q) \\
&= P_\pi(p-q)(s^*, a^*) - P_\pi(p-q)(s_*, a_*) \\
&\leq \max_{(s,a)\in\mathcal{S}\times\mathcal{A}} P_\pi(p-q)(s,a) - \min_{(s,a)\in\mathcal{S}\times\mathcal{A}} P_\pi(p-q)(s,a) \\
&= \|P_\pi(p-q)\|_{sp,\infty}
\end{aligned} \tag{82}$$

Denote $\tau := (s,a)$. Note that

$$\|P_\pi(p-q)\|_{sp,\infty} = \max\left\{\sum_\tau P_\pi(\tau|\tau_1)(p-q)(\tau_1) - \sum_\tau P_\pi(\tau|\tau_2)(p-q)(\tau_2)\right\} \tag{83}$$

So it is sufficient to prove that

$$\|P_\pi(p-q)\|_{sp,\infty} \leq \gamma \|p-q\|_{sp,\infty}. \tag{84}$$

For notational convenience, let $b(\tau_1, \tau_2|\tau) = \min\{P_\pi(\tau|\tau_1), P_\pi(\tau|\tau_2)\}$. We prove the above inequality by first showing that the inequality holds for any $\tau := (s_1, a_1)$, and then taking a maximum over all $(s,a)$ pairs.

$$\begin{aligned}
&\sum_\tau P_\pi(\tau|\tau_1)(p-q)(\tau_1) - \sum_\tau P_\pi(\tau|\tau_2)(p-q)(\tau_2) \\
&= \sum_\tau [P_\pi(\tau|\tau_1) - b(\tau_1,\tau_2|\tau)](p-q)(\tau_1) - \sum_\tau [P_\pi(\tau|\tau_2) - b(\tau_1,\tau_2|\tau)](p-q)(\tau_2) \\
&\leq \sum_\tau [P_\pi(\tau|\tau_1) - b(\tau_1,\tau_2|\tau)]\max(p-q) - \sum_\tau [P_\pi(\tau|\tau_2) - b(\tau_1,\tau_2|\tau)]\min(p-q) \\
&= \sum_\tau [P_\pi(\tau|\tau_1) - b(\tau_1,\tau_2|\tau)](p-q)(s^*, a^*) - \sum_\tau [P_\pi(\tau|\tau_2) - b(\tau_1,\tau_2|\tau)](p-q)(s_*, a_*) \\
&= [1 - \sum_\tau b(\tau_1,\tau_2;\tau)]\|p-q\|_{sp,\infty} \tag{85} \\
&\leq \gamma \|p-q\|_{sp,\infty} \tag{86}
\end{aligned}$$

Simply taking the maximum over all state-action pairs we can get the desired result. $\square$

In view of A.6, when the MDP does not have 1 step contraction, we can construct a $J$-step Bellman operator for some integer $J$, which would be sufficient due to Assumption 2.1 (see Theorem 8.5.3 [27]).

Now we are ready to prove Theorem 4.5.

*Proof.* Taking expectation w.r.t $\zeta_k$ and using triangle inequality

$$\left\|\mathbb{E}_{\zeta_k}\bar{\mathcal{Q}}_{\theta_k}^{\pi_k,\zeta_k} - \bar{Q}_{\theta_k}^{\pi_{\theta_k}}\right\|_{sp,\infty} \leq \left\|\bar{Q}_{\theta_k}^{\pi_k} - \bar{Q}_{\theta_k}^{\pi_{\theta_k}}\right\|_{sp,\infty} + \left\|\mathbb{E}_{\zeta_k}\bar{\mathcal{Q}}_{\theta_k}^{\pi_k,\zeta_k} - \bar{Q}_{\theta_k}^{\pi_k}\right\|_{sp,\infty},$$

$$\leq \left\|\bar{Q}_{\theta_k}^{\pi_k} - \bar{Q}_{\theta_k}^{\pi_{\theta_k}}\right\|_{sp,\infty} + \varsigma. \tag{87}$$

Here $\bar{Q}_{\theta_k}^{\pi_k}$ is the ideal differential $Q$-function in the $k$th iteration, and $\bar{Q}_{\theta_k}^{\pi_{\theta_k}}$ is the optimal differential $Q$-function w.r.t to the current reward function $c(s,a;\theta_k)$. We cannot expect them to be close from the beginning, as we only partially solve the entropy regularized RL problem. We show in this theorem that the gap between them is shrinking.

We denote $\bar{Q}_{\theta}^{\pi}$ as $\bar{Q}^{\pi}(s,a;\theta)$ for clarity, as it represents the differential $Q$-function with the reward function parameterized by $\theta$ and follow sample distribution induced by policy $\pi$.

Note that

$$\left\|\bar{Q}^{\pi_k}(s,a;\theta_k) - \bar{Q}^{\pi_{\theta_k}}(s,a;\theta_k)\right\|_{sp,\infty}$$
$$= \left\|\bar{Q}^{\pi_k}(s,a;\theta_k) - \bar{Q}^{\pi_{\theta_k}}(s,a;\theta_k) + \bar{Q}^{\pi_{\theta_{k-1}}}(s,a;\theta_{k-1}) - \bar{Q}^{\pi_{\theta_{k-1}}}(s,a;\theta_{k-1})\right.$$
$$\left. + \bar{Q}^{\pi_k}(s,a;\theta_{k-1}) - \bar{Q}^{\pi_k}(s,a;\theta_{k-1})\right\|_{sp,\infty}$$
$$\leq \left\|\bar{Q}^{\pi_{\theta_{k-1}}}(s,a;\theta_{k-1}) - \bar{Q}^{\pi_{\theta_k}}(s,a;\theta_k)\right\|_{sp,\infty} + \left\|\bar{Q}^{\pi_k}(s,a;\theta_k) - \bar{Q}^{\pi_k}(s,a;\theta_{k-1})\right\|_{sp,\infty}$$
$$+ \left\|\bar{Q}^{\pi_{\theta_{k-1}}}(s,a;\theta_{k-1}) - \bar{Q}^{\pi_k}(s,a;\theta_{k-1})\right\|_{sp,\infty}$$
$$\overset{(i)}{\leq} L_q\left\|\theta_k - \theta_{k-1}\right\|_2 + L_q\left\|\theta_k - \theta_{k-1}\right\|_2 + \left\|\bar{Q}^{\pi_{\theta_{k-1}}}(s,a;\theta_{k-1}) - \bar{Q}^{\pi_{k-1}}(s,a;\theta_{k-1})\right\|_{sp,\infty}. \tag{88}$$

where $(i)$ follows Lemma A.4-A.5. Note that from the update rule $\theta_k = \theta_{k-1} - \alpha g_k$,

$$\left\|\theta_k - \theta_{k-1}\right\|_2 \leq \alpha\left\|g_k\right\|_2 = \alpha\left\|\nabla[\rho_E - \rho_{\pi_k}]\right\|_2 \leq \alpha L_c \tag{89}$$

For the second term, note that

$$\bar{Q}^{\pi_{\theta_{k-1}}}(s,a;\theta_{k-1}) = \mathcal{T}\bar{Q}^{\pi_{\theta_{k-1}}}(s,a;\theta_{k-1}), \tag{90}$$

$$\bar{Q}^{\pi_k}(s,a;\theta_{k-1}) = \mathcal{T}\bar{Q}^{\pi_{k-1}}(s,a;\theta_{k-1}). \tag{91}$$

From lemma A.6 we have

$$\left\|\bar{Q}^{\pi_{\theta_{k-1}}}(s,a;\theta_{k-1}) - \bar{Q}^{\pi_{k-1}}(s,a;\theta_{k-1})\right\|_{sp,\infty} \leq \gamma\left\|\bar{Q}_{\theta_{k-1}}^{\pi_{k-1}} - \bar{Q}_{\theta_{k-1}}^{\pi_{\theta_{k-1}}}\right\|_{sp,\infty}. \tag{92}$$

Combine the above inequalities

$$\left\|\mathbb{E}_{\zeta_k}\bar{\mathcal{Q}}_{\theta_k}^{\pi_k,\zeta_k} - \bar{Q}_{\theta_k}^{\pi_{\theta_k}}\right\|_{sp,\infty} \leq \gamma\left\|\bar{Q}_{\theta_{k-1}}^{\pi_{k-1}} - \bar{Q}_{\theta_{k-1}}^{\pi_{\theta_{k-1}}}\right\|_{sp,\infty} + 2\alpha L_q L_c + \varsigma. \tag{93}$$

Sum from $k = 1$ to $K$, rearrange the equations, we have

$$\frac{1}{K}\sum_{k=0}^{K-1}\left\|\bar{\mathcal{Q}}_{\theta_k}^{\pi_k,\zeta_k} - \bar{Q}_{\theta_k}^{\pi_{\theta_k}}\right\|_{sp,\infty} \leq \frac{\left\|\bar{Q}_{\theta_0}^{\pi_0} - \bar{Q}_{\theta_0}^{\pi_{\theta_0}}\right\|_{sp,\infty}}{(1-\gamma)K} + \frac{2\alpha L_q L_c + \varsigma}{1-\gamma}. \tag{94}$$

Pick step size $\alpha = \frac{\alpha_0}{\sqrt{K}}$ with $\alpha_0 > 0$, we have

$$\frac{1}{K}\sum_{k=0}^{K-1}\left\|\bar{\mathcal{Q}}_{\theta_k}^{\pi_k,\zeta_k} - \bar{Q}_{\theta_k}^{\pi_{\theta_k}}\right\|_{sp,\infty} = \mathcal{O}(K^{-1}) + \mathcal{O}(K^{-1/2}) + \frac{\varsigma}{1-\gamma}. \tag{95}$$

$\square$

## A.8  Proof of Theorem 4.6

*Proof.* Note that a constant shift over the action-value function induces the same policy. It is easy to show the following relation as presented in [37]

$$\|\mathbb{E}_{\zeta_k}\log\pi_{k+1} - \log\pi_{\theta_k}\|_\infty \le \|\bar{\mathcal{Q}}^{\pi_k,\zeta_k} - \bar{Q}^{\pi_k}_{\theta_k}\|_\infty$$
$$= \|\bar{\mathcal{Q}}^{\pi_k,\zeta_k} - \bar{Q}^{\pi_k}_{\theta_k} + c_{\theta_k}e - c_k e\|_\infty. \tag{96}$$

The relation holds for any $c_{\theta_k}, c_k$. Denoting $c = c_{\theta_k} - c_k$, we conclude that

$$\|\mathbb{E}_{\zeta_k}\log\pi_{k+1} - \log\pi_{\theta_k}\|_\infty \le \min_{c\in\mathbb{R}}\|\mathbb{E}_{\zeta_k}\bar{\mathcal{Q}}^{\pi_k,\zeta_k} - \bar{Q}^{\pi_k}_{\theta_k} + ce\|_\infty = \tfrac{1}{2}\left\|\mathbb{E}_{\zeta_k}\bar{\mathcal{Q}}^{\pi_k,\zeta_k} - \bar{Q}^{\pi_k}_{\theta_k}\right\|_{sp,\infty}. \tag{97}$$

In view of Theorem 4.6, we have

$$\tfrac{1}{K}\sum_{k=0}^{K-1}\|\mathbb{E}_{\zeta_k}[\log\pi_{k+1}] - \log\pi_{\theta_k}\|_\infty \le \frac{\left\|\bar{Q}^{\pi_0}_{\theta_0} - \bar{Q}^{\pi_{\theta_0}}_{\theta_0}\right\|_{sp,\infty}}{2(1-\gamma)K} + \frac{\alpha L_q L_c + \varsigma}{2(1-\gamma)}. \tag{98}$$

The convergence result for reward function approximation is similar to smooth-nonconvex problems in optimization. We begin by using the smoothness property of the objective function $L$.

$$L(\theta_{k+1}) \le L(\theta_k) + \langle\nabla_\theta L(\theta_k), \theta_{k+1} - \theta_k\rangle + \tfrac{L_c}{2}\|\theta_{k+1} - \theta_k\|_2^2$$
$$\overset{(i)}{=} L(\theta_k) - \alpha\langle\nabla_\theta L(\theta_k), g_k\rangle - \tfrac{1}{2}\alpha^2 L_c\|g_k\|_2^2$$
$$\le L(\theta_k) - \alpha\langle\nabla_\theta L(\theta_k), g_k - \nabla_\theta L(\theta_k)\rangle - \alpha\|\nabla_\theta L(\theta_k)\|_2^2 + \tfrac{1}{2}\alpha^2 L_c\|g_k\|_2^2$$
$$\overset{(ii)}{=} L(\theta_k) - \alpha\langle\nabla_\theta L(\theta_k), g_k - \nabla_\theta L(\theta_k)\rangle - \alpha\|\nabla_\theta L(\theta_k)\|_2^2 + 2\alpha^2 L_c^3. \tag{99}$$

where $(i)$ follows the update rule; $(ii)$ follows the bound on the gradient $g_k$, i.e.

$$\|g_k\|_2 \le \|\mathbb{E}_{(s,a)\sim d^E}[\nabla_\theta c(s,a;\theta_k)] - \mathbb{E}_{(s,a)\sim d^\pi}[\nabla_\theta c(s,a;\theta_k)]\|_2$$
$$\le \|\mathbb{E}_{(s,a)\sim d^E}[\nabla_\theta c(s,a;\theta_k)]\|_2 + \|\mathbb{E}_{(s,a)\sim d^\pi}[\nabla_\theta c(s,a;\theta_k)]\|_2$$
$$\le \max\|\nabla_\theta c\|_2 + \max\|\nabla_\theta c\|_2$$
$$= 2L_c. \tag{100}$$

Taking expectation w.r.t $\zeta_k$ and state-action sample distribution $d^{\theta_k}$, on the above equation,

$$\mathbb{E}[L(\theta_{k+1})] \le \mathbb{E}[L(\theta_k)] - \alpha\mathbb{E}[\langle\nabla_\theta L(\theta_k), g_k - \nabla_\theta L(\theta_k)\rangle] - \alpha\mathbb{E}[\|\nabla_\theta L(\theta_k)\|_2^2] + 2\alpha^2 L_c^3$$
$$= \mathbb{E}[L(\theta_k)] - \alpha\mathbb{E}[\langle\nabla_\theta L(\theta_k), \mathbb{E}[g_k - \nabla_\theta L(\theta_k)|\theta_k]\rangle] - \alpha\mathbb{E}[\|\nabla_\theta L(\theta_k)\|_2^2] + 2\alpha^2 L_c^3$$
$$\overset{(i)}{\le} \mathbb{E}[L(\theta_k)] + 2\alpha L_c\mathbb{E}\left[\left\|\mathbb{E}_{(s,a)\sim d^{\pi_{k+1}}}[\nabla_\theta c(s,a;\theta_k)] - \mathbb{E}_{(s,a)\sim d^{\pi_{\theta_k}}}[\nabla_\theta c(s,a;\theta_k)]\right\|\right]$$
$$\quad - \alpha\mathbb{E}[\|\nabla_\theta L(\theta_k)\|_2^2] + 2\alpha^2 L_c^3$$
$$= \mathbb{E}[L(\theta_k)] + 4\alpha L_c\|\nabla_\theta c\|_2\mathbb{E}[\|d(s,a;\pi_{\theta_k}) - d(s,a;\pi_{k+1})\|_{TV}]$$
$$\quad - \alpha\mathbb{E}[\|\nabla_\theta L(\theta_k)\|^2] + 2\alpha^2 L_c^3$$
$$\overset{(ii)}{\le} \mathbb{E}[L(\theta_k)] + 4\alpha L_c\max\|\nabla_\theta r\|_2\mathbb{E}[\|d(s,a;\pi_{\theta_k}) - d(s,a;\pi_{k+1})\|_{TV}]$$
$$\quad - \alpha\mathbb{E}[\|\nabla_\theta L(\theta_k)\|^2] + 2\alpha^2 L_c^3$$
$$\overset{(iii)}{\le} \mathbb{E}[L(\theta_k)] + 4\alpha L_c L_r\mathbb{E}[\|d(s,a;\pi_{\theta_k}) - d(s,a;\pi_{k+1})\|_{TV}]$$
$$\quad - \alpha\mathbb{E}[\|\nabla_\theta L(\theta_k)\|^2] + 2\alpha^2 L_c^3$$
$$\overset{(iv)}{\le} \mathbb{E}[L(\theta_k)] + 4\alpha L_c L_r C_d\sqrt{|\mathcal{S}|\cdot|\mathcal{A}|}\mathbb{E}\left[\left\|\bar{\mathcal{Q}}^{\pi_{\theta_k}}(s,a;\theta_k) - \bar{Q}^{\pi_{k+1}}(s,a;\theta_k) + ce\right\|_\infty\right],$$
$$\quad - \alpha\mathbb{E}[\|\nabla_\theta L(\theta_k)\|^2] + 2\alpha^2 L_c^3$$
$$\le \mathbb{E}[L(\theta_k)] + 2\alpha L_c L_r C_d\sqrt{|\mathcal{S}|\cdot|\mathcal{A}|}\mathbb{E}\left[\left\|\bar{\mathcal{Q}}^{\pi_{\theta_k},\zeta_k}_{\theta_k} - \bar{Q}^{\pi_k}_{\theta_k}\right\|_{sp,\infty}\right]$$
$$\quad - \alpha\mathbb{E}[\|\nabla_\theta L(\theta_k)\|^2] + 2\alpha^2 L_c^3. \tag{101}$$

where $(i)$ follows Cauchy-Schwartz inequality, notice that

$$
\begin{aligned}
&- \alpha\mathbb{E}\left[\langle\nabla_\theta L(\theta_k), \mathbb{E}\left[g_k - \nabla_\theta L(\theta_k)|\theta_k\right]\rangle\right] \\
&= \alpha\mathbb{E}\left[\langle\nabla_\theta L(\theta_k), \mathbb{E}\left[-g_k + \nabla_\theta L(\theta_k)|\theta_k\right]\rangle\right] \\
&\leq \alpha\|\nabla_\theta L(\theta_k)\|\mathbb{E}\left[\left\|\mathbb{E}_{(s,a)\sim d^{\pi_{k+1}}}\left[\nabla_\theta c(s,a;\theta_k)\right] - \mathbb{E}_{(s,a)\sim d^{\pi_{\theta_k}}}\left[\nabla_\theta c(s,a;\theta_k)\right]\right\|\right].
\end{aligned}
\tag{102}
$$

and that

$$
\begin{aligned}
&\mathbb{E}\left[\left\|\mathbb{E}_{(s,a)\sim d^{\pi_{k+1}}}\left[\nabla_\theta c(s,a;\theta_k)\right] - \mathbb{E}_{(s,a)\sim d^{\pi_{\theta_k}}}\left[\nabla_\theta c(s,a;\theta_k)\right]\right\|\right] \\
&\leq 2\|\nabla_\theta c\|_2\mathbb{E}\left[\|d(s,a;\pi_{\theta_k}) - d(s,a;\pi_{k+1})\|_{TV}\right].
\end{aligned}
\tag{103}
$$

follows the Hölder's inequality; $(ii)$ follows the definition of the total-variation norm; $(iii)$ follows the bound on the gradient $\nabla_\theta c$; $(iv)$ follows from the fact that the policy is the Boltzmann distribution of the exponential of the action-value function, Lemma A.3, and equivalence of norms. Rearrange terms we have

$$
\begin{aligned}
\alpha\mathbb{E}\left[\|\nabla_\theta L(\theta_k)\|_2^2\right] \leq{}& \mathbb{E}\left[L(\theta_k)\right] - \mathbb{E}\left[L(\theta_{k+1})\right] + 2L_c^3\alpha^2 \\
&+ 2\alpha L_c L_r C_d\sqrt{|\mathcal{S}|\cdot|\mathcal{A}|}\mathbb{E}\left[\left\|\bar{\mathcal{Q}}_{\theta_k}^{\pi_{\theta_k},\zeta_k} - \bar{Q}_{\theta_k}^{\pi_k}\right\|_{sp,\infty}\right].
\end{aligned}
\tag{104}
$$

Sum from $k = 0$ to $K - 1$, and divide both sides by $\alpha K$,

$$
\begin{aligned}
\frac{1}{K}\sum_{k=0}^{K-1}\mathbb{E}\left[\|\nabla_\theta L(\theta_k)\|_2^2\right] \leq{}& 2L_c^3\alpha + \frac{\mathbb{E}[L(\theta_0)] - \mathbb{E}[L(\theta_K)]}{\alpha K} \\
&+ \frac{2L_c L_r C_d\sqrt{|\mathcal{S}|\cdot|\mathcal{A}|}}{K}\sum_{k=0}^{K-1}\mathbb{E}\left[\left\|\bar{\mathcal{Q}}_{\theta_k}^{\pi_{\theta_k},\zeta_k} - \bar{Q}_{\theta_k}^{\pi_k}\right\|_{sp,\infty}\right].
\end{aligned}
\tag{105}
$$

In view of theorem 4.5, we have

$$
\begin{aligned}
&\frac{1}{K}\sum_{k=0}^{K-1}\mathbb{E}\left[\|\nabla_\theta L(\theta_k)\|_2^2\right] \\
&\leq 2L_c^3\alpha + \frac{\mathbb{E}[L(\theta_0)] - \mathbb{E}[L(\theta_K)]}{\alpha K} + \frac{2L_c L_r C_d\sqrt{|\mathcal{S}|\cdot|\mathcal{A}|}}{K}\sum_{k=0}^{K-1}\mathbb{E}\left[\left\|\bar{\mathcal{Q}}_{\theta_k}^{\pi_{\theta_k},\zeta_k} - \bar{Q}_{\theta_k}^{\pi_k}\right\|_{sp,\infty}\right] \\
&\leq 2L_c^3\alpha_0/\sqrt{K} + \alpha_0\frac{\mathbb{E}[L(\theta_0)] - \mathbb{E}[L(\theta_K)]}{\sqrt{K}} + \frac{2L_c L_r C_d\sqrt{|\mathcal{S}|\cdot|\mathcal{A}|}\left\|\bar{\mathcal{Q}}_{\theta_0}^{\pi_0} - \bar{Q}_{\theta_0}^{\pi_{\theta_0}}\right\|_{sp,\infty}}{(1-\gamma)K} \\
&\quad+ \frac{4\alpha_0 L_q L_c^2 L_r C_d\sqrt{|\mathcal{S}|\cdot|\mathcal{A}|}}{\sqrt{K}(1-\gamma)} + \frac{2\nu L_c L_r C_d\sqrt{|\mathcal{S}|\cdot|\mathcal{A}|}}{1-\gamma} \\
&= \mathcal{O}(K^{-1}) + \mathcal{O}(K^{-1/2}) + \frac{2\nu L_c L_r C_d\sqrt{|\mathcal{S}|\cdot|\mathcal{A}|}}{1-\gamma}.
\end{aligned}
\tag{106}
$$

Note that

$$
\begin{aligned}
\mathbb{E}\left[\left\|\bar{\mathcal{Q}}_{\theta_k}^{\pi_{\theta_k},\zeta_k} - \bar{Q}_{\theta_k}^{\pi_k}\right\|_{sp,\infty}\right] &= \mathbb{E}\left[\left\|\bar{\mathcal{Q}}_{\theta_k}^{\pi_k,\zeta_k} - \bar{Q}_{\theta_k}^{\pi_{\theta_k}} + \bar{Q}_{\theta_k}^{\pi_{\theta_k}} - \bar{Q}_{\theta_k}^{\pi_k}\right\|_{sp,\infty}\right] \\
&\leq \mathbb{E}\left[\left\|\bar{Q}_{\theta_k}^{\pi_{\theta_k}} - \bar{Q}_{\theta_k}^{\pi_k}\right\|_{sp,\infty}\right] + \mathbb{E}\left[\left\|\bar{\mathcal{Q}}_{\theta_k}^{\pi_{\theta_k},\zeta_k} - \bar{Q}_{\theta_k}^{\pi_k}\right\|_{sp,\infty}\right] \\
&\leq \mathbb{E}\left[\left\|\bar{Q}_{\theta_k}^{\pi_{\theta_k}} - \bar{Q}_{\theta_k}^{\pi_k}\right\|_{sp,\infty}\right] + \nu.
\end{aligned}
\tag{107}
$$

$\square$

## A.9 Experiment Details

Here we present some details about the experiments. All experiments are performed on an Apple MacBook Pro with an M1 chip. For RL tasks, We use a discount factor $\gamma = 0.99$ for training the SAC. SAC and SPMD agents are trained with $3e6$ samples, with a learning rate of $3e - 4$. The average-reward estimate is calculated by taking an average of the sampled batch. The training process of each agent takes around 3 hours for one environment.

For IRL tasks, the expert demonstrations are collected from training a SAC agent for five million steps. We train the IPMD agent $2e6$ samples (number of interactions with the environment) while training IQ-Learn and $f$-IRL using the script provided along with its implementation. To avoid the objective from exploding, we also add a scaled (0.05 the size of) norm of the predicted reward as a regularization term. The inexact evaluation of the gradient can be incorporated into the error term in stochastic gradient descent so that this change has no major impact on the analysis.

For the reward recovery experiment, we train the expert agent using SAC with a 0.99 discount factor. IQ-Learn is trained with 1e5 iterations for Pendulum and 5e5 iterations for LunarLanderContinuous. IPMD is trained with 1e5 iterations for both environments. Both agents use 11 expert trajectories.

## A.10 Miscellaneous

We acknowledge that IRL, like many other machine learning techniques, has potential implications if misused. IRL can be used in violation of privacy (as the reviewer mentioned) by inferring an individual's intentions and preferences, potentially crafting convincing social engineering attacks or phishing attempts; IRL can also be used to model the behavior of specific demographics, which could result in biased algorithmic decision-making, leading to unfair treatment or discrimination against certain groups, etc.