# OpenReview forum: "Inverse Reinforcement Learning with the Average Reward Criterion"
_NeurIPS.cc/2023/Conference — NeurIPS 2023 poster_

### Official Review · Reviewer_DtTS · 2023-07-06

**Soundness:** 3 good
**Presentation:** 3 good
**Contribution:** 3 good
**Rating:** 6
**Confidence:** 3

**Summary:**

This paper addresses Inverse Reinforcement Learning (IRL) in the average-reward setting. The authors propose a Stochastic Policy Mirror Descent (SPMD) method to solve the Average Reward Markov Decision Process subproblem and use SPMD to propose the Inverse Policy Mirror Descent method to solve the IRL problem. The authors provide complexity results for these methods and experimentally validate them using the MuJoCo benchmark.

**Strengths:**

- The authors develop a novel method and show its merits both theoretically and experimentally compared to existing methods.
- The theoretical analysis is very thorough and the authors clearly outline the assumptions that are made.
- The proposed IPMD method achieves good results on a variety of environments.

**Weaknesses:**

- The paper is a bit light on experiment details (e.g. hyperparameters). More detail here would be appreciated (e.g. see Questions).
- It’d be nice if the authors could provide an intuitive explanation for the result of Theorem 3.4. Equation 25 is a bit difficult to understand.
- The authors make a fair number of assumptions in their theoretical analysis. It'd be nice if the authors added some discussion on which assumptions hold in practice and/or how they enforce those assumptions, as well as how restrictive these assumptions are.

**Questions:**

- Does Assumption 3.2 hold in practice? Do the authors do anything to enforce these constraints?
- What is $\omega$ in Equation (3) and Algorithm 1 chosen to be in practice?
- Can the authors provide architecture details for the policy, Q network, and learned reward function in the appendix?
- Equation 24 in Theorem 3.4 makes an assumption about step size. Is this assumption implemented in practice (i.e. in Algorithm 1 what is $\eta_k$ chosen to be?
- It seems the authors assume access to a large number of expert trajectories (Appendix A.9 says expert demonstrations are collected for five million steps). I’m curious how IPMD would perform with a more limited number of expert trajectories (for instance, the f-IRL[1] paper has experiments where they just use 1 expert demonstration). Have the authors considered how number of expert trajectories affects performance of IPMD?

[1] f-IRL: Inverse Reinforcement Learning via State Marginal Matching (Ni et al.)

**Limitations:**

yes

---

> ### Author Rebuttal · Authors · 2023-08-08
>
> We extend our sincere appreciation for your insightful review and thoughtful comments, which greatly contribute to the refinement of our paper. We are pleased to address each of your points below:
>
> 1. We recognize the importance of providing comprehensive experiment details, including hyperparameters, to enhance the transparency and reproducibility of our work. In response to your suggestion, we will furnish a more comprehensive account of hyperparameters, encompassing aspects such as learning rates, discount factors, and neural network architectures. Please see our response to later questions and our global reply.
>
> 2. We value your input regarding Equation 25 and its comprehension. Equation 25 encapsulates a convergence overview without specifying explicit step size choices. Such formulations, common in optimization literature, highlight the significance of step size selection on convergence speed. To be specific, $\alpha_k, \beta_k$ are both step sizes and the central term $\sum \rho_k - \rho^*$ (distance to the optimal value) plus $D(\pi_k, \pi^*)$ (the distance of the current policy to the optimal policy) will shrink as $k$ grows, which is be bounded by some combination of $\alpha_k$and $\beta_k$ plus an irreducible function approximation error $\varsigma$. The left-hand side is a weighted average of function value error plus an average error from step $k=1$ to $k=K$. We acknowledge the less intuitive nature of Theorem 3.4 and will provide a more intuitive explanation of its implications, coupled with illustrative examples, to enhance clarity. For a specific step size choice and the condition on $\mu$, we get different rates of convergence, which is presented in Corollary 3.5.
>
> 3. Assumption 2.1 (uniform ergodicity) is considered restrictive but often necessary for analysis, albeit potential violations in practice. Assumption 3.2's practicality hinges on model choices; if neural networks are employed, gradient "clipping" may be utilized. This also applies to Assumptions 4.1 and 4.2. Assumptions 3.3, 3.6, and 4.3, concerning stochastic estimator proximity, are less restrictive given bounded state and action spaces.  Your insight into detecting issues in practice is well-taken, and we will incorporate a comprehensive discussion to highlight these nuances.
> In practice, it is usually obvious when things go wrong, e.g., when the differential Q-function explodes to 10e6. This can happen due to rare occasions and the culprit is usually detectable, e.g., a large learning rate. Please refer to our overall response for additional discussion.
>
> 4. As elucidated in Section 3.1, in practice, we select $h$ and $\omega$ as negative entropy due to the subproblem's entropy-regularized nature. However, alternative distance-generating functions for Bregman distance beyond KL divergence are possible, and PMD can accommodate different choices. We will provide additional clarification on this choice and its implications.
>
> 5. The policy network employs two fully connected hidden layers of dimension 256 each, taking actions as input and outputting a distribution. Both the Q network and reward function share the same architecture, with ReLU activation used in hidden layers. A double Q-learning technique minimizes overestimation. We will incorporate comprehensive architecture specifications in the appendix for reference.
>
> 6. We appreciate your inquiry into the implementation of the assumption in Equation 24. Our comparative exploration of various step size schedules yielded minimal differences, possibly due to subproblem optimization via a few gradient descent steps. We will emphasize this aspect's relevance and practical implementation.
>
> 7. Your query on IPMD's performance sensitivity to expert trajectory count is insightful. While unreported at submission, IPMD exhibits strong performance on locomotion tasks involving complex robots with a sole expert demonstration. For example, in one of our ongoing projects, we test IPMD on the MuJoCo Cassie robot (https://mujoco.readthedocs.io/en/stable/models.html), hope Cassie can walk on random terrain from demonstrations. IPMD can reach episodic reward 479.815 from one expert demonstration of episodic reward 447.1955, outperforming the expert demonstration. This suggests IPMD's efficiency in utilizing limited expert demonstrations.
>
> Once again, we thank you for your comprehensive review and look forward to addressing your valuable feedback in our revisions.

---

> > ### Comment · Reviewer_DtTS · 2023-08-18
> >
> > Thank you for the clarifications!

---

### Official Review · Reviewer_ifE7 · 2023-07-06

**Soundness:** 4 excellent
**Presentation:** 3 good
**Contribution:** 2 fair
**Rating:** 3
**Confidence:** 4

**Summary:**

In this paper inverse reinforcement learning is studied when the teacher was using an average-reward criterion rather than discounted rewards with known discount factor. The paper proposes a stochastic first-order method starting from stochastic policy mirror descent for MDPs and continuing towards inverse policy mirror descent for solving the IRL problem. The paper contains also some numerical experiments based on MuJoCo benchmarks.


**Strengths:**

The paper is very clear and elegant in its use of average reward method in RL and IRL. There are many papers on duality of this type although not yet specifically on the average reward criterion, although in NAC papers similar considerations were made.

The supplementary material is strictly supplementary and very clear and comprehensive.

It's great the hints for practical use are included.

**Weaknesses:**

“To the best of our knowledge” is not a useful sentence in an abstract, in particular in an anonymous manuscript. It is more useful to discuss such opinions later in the paper in the context of related results (see e.g. previous NeurIPS conferences).

Duality has been used in many similar contexts, but it could still be said more clearly why it is a good idea here. It is acceptable, if you want to avoid discussing natural AC, but then there is more to explain.

Experiments are relatively few with just final results given, so no understanding why the proposed method is useful can be gained. The is no discussion of the experimental results, therefore, although one may guess, it is not clear why the proposed method struggles on Ant or whether there is any relation of the performances comparing tables 1 and 2. There is no attempt included to check whether the complexity results are tight or whether global optima are indeed found.

Check: “Acrot”

Assumptions 4.2 and 4.3 are formally stated in the main text, but do not seem to be used. In the appendix only 4.2 is mentioned.
Ref. 28 is not mentioned in the text, perhaps a separate bibliography for the appendix would make sense.

I find it a bit difficult to distinguish between all the Qs. The distinction between math-Q and cal-Q is probably necessary, For example, tilde-cal-Q (32) could as well be substituted by its definition. It occurs again only in (34) and (38), where at least a ref to the definition needs to be included. In “differential Q-function” (in italics in the main text vs. theorem environment) different fonts are used, which should be avoid this, ideally by using the math-Q font that seems to occur also in the theorem environment.

**Questions:**

Can you explain more about the improvements (as in Humanoid RL) and deficiencies (as in Ant IRL or walker RL)? What feature of the algorithm contributes to the success or suboptimality of the performance in each problem? Are the assumption not strictly satisfied, or is the sampling the reason for any suboptimality?

Which theorems have appeared in earlier work or have strong similarities to theorems in earlier work?

I would assume that 1/(1-gamma) not much smaller than K, and the difference between them is related to the number of states and actions. In other words, is there a practical need for the O-symbols or is the complexity already implied by the known factor proportional to 1/(1-gamma)?

Why does the performance show variablity even for simple problems (see Fig. 1)?


**Limitations:**

The authors state in the supplementary material: "We anticipate no potential negative societal impacts concerning this research." which could seem questionable in the context of IRL, because IRL has the potential to uncover hidden motives in legal human behavior without the consent of a person. The research seems fine, but some discussion, beyond the mere statement would be useful.

---

> ### Author Rebuttal · Authors · 2023-08-08
>
> Thank you for the thorough review and insightful comments. We appreciate your time spent on the review and detailed comments on the paper. First we want to point out that our paper focuses on the theoretical development of solving the Inverse Reinforcement Learning (IRL) problem under the average-reward setting. Note that almost all analyses on the discounted MDPs highly rely on (some form of) the contraction property given by the discount factor, and their work cannot be directly translated into the average-reward setting.
>
> 1. Regarding "To the best of our knowledge" in the abstract, we acknowledge your point and will consider rephrasing it in subsequent revisions.
>
> 2. We recognize the need for a clearer justification of the use of duality. In the context of Inverse Reinforcement Learning (IRL), we highlight that directly solving the primal problem presents challenges, including maintaining constraints and ensuring maximum entropy policies, which are not resolved effectively in the community. Exploiting duality has a history in IRL, allowing us to leverage a more structured dual problem, enhancing our ability to devise effective optimization techniques. Regarding using Natural Actor-Critic (NAC), it is true that one may solve the subproblem with NAC. But there's no existing theoretical analysis on the rate of convergence for NAC for solving the average-reward Markov Decision Processes (AMDPs). As mentioned in the paper, [Cen 2021] uses NAC to solve a discounted MDP with a finite state and action space. Whether its method and analysis can be adapted to solve our problem (average reward with general state and action space and general function approximation) needs nontrivial analysis.
>
> Shicong Cen, Chen Cheng, Yuxin Chen, Yuting Wei, and Yuejie Chi. Fast global convergence
> 314 of natural policy gradient methods with entropy regularization. Operations Research, 2021.
>
> 3. While we acknowledge the relatively limited number of experiments and the absence of an in-depth discussion of results, our primary focus was on the theoretical underpinnings of our proposed methods. The experiments validate our theoretical findings and illustrate competitive or superior performance, even without exhaustive exploration of hyperparameters. Notice that since the performance metric is the episodic return, methods that work with discounted MDPs are approximately solving the AMDP, and the performance is very sensitive to the discount factor. In IRL, guessing the discount factor is crucial to estimate the reward. Our method can naturally avoid these problems.
>
> 4. We understand your point about complexity analysis and global optima. Given the complexities introduced by function approximation, proving tight complexity bounds or identifying global optima remains challenging. Notice that there are active efforts in the theoretical community hoping to derive a "tight" complexity bound for general Reinforcement Learning, yet no general results are obtained. Additionally, in the context of non-convex optimization, a global optimum is usually unattainable. Nevertheless, our approach offers insights into practical reward recovery, as demonstrated in the reward recovery experiment comparing our method to Inverse Q-learning (IQL), a discounted method. We show that even with the correct discount factor when using IQL, our method is 10x times more accurate.
>
> 5. We appreciate your observation about Assumptions 4.2 and 4.3. Assumption 4.2 is crucial in our analysis framework, impacting Lemma A.4 (Equation 72), A.5 (Equation 78), and Thm 4.5 (Equation 88), while Assumption 4.3 is essential for Thm 4.5 (in Equation 93) and 4.6 (Equation 104-107).
>
> 6. We understand your concern about distinguishing between different Q-functions. Our choice of $\tilde{\mathcal{Q}}$ aims for conciseness, though we will ensure better clarity in referencing definitions. We will consider using more distinct notations for denoting the differential $Q$-function and its stochastic estimator.
>
> 7. We acknowledge the request for a deeper discussion of algorithmic improvements and deficiencies in various environments. For a detailed analysis of the algorithm, please refer to our overall reply.
>
> 8. We clarify that our theoretical contributions leverage convex/non-convex optimization techniques. Our formulation of average-reward IRL is novel, bridging the gap between the current understanding of average-reward Markov Decision Processes (AMDPs) and IRL under such settings. While some theorems may appear similar to existing work, adapting them to this novel context requires nontrivial analysis, as evidenced by the cited lemmas.
>
> 9. In our complexity analysis, the $O$ notation captures convergence rates, while constants such as $1/(1-\gamma)$ play lesser importance as $K$ can grow sufficiently large. This notation is widely used to convey the speed of convergence. It provides a concise and informative way to describe the growth rate of a function relative to a specific parameter (usually the problem size or the number of iterations).
>
> 10. The observed variability in performance, even for simple problems, can be attributed to the stochastic nature of the environments. When an environment is initialized, a random noise is added by the simulator. That contributes to the variability
>
> 11. Your ethical concern is valid, and we share your perspective on the responsible use of IRL. Please refer to our overall response for a detailed discussion.
>
> Once again, we thank you for your comprehensive review and look forward to addressing your valuable feedback in our revisions.

---

> > ### Comment · Reviewer_ifE7 · 2023-08-21
> >
> > Thank you for the comprehensive reply to my concerns, which confirms, on the one hand, that the points I have made were mostly valid, but, on the other hand, confirm that there is clearly the potential to improve the paper, although there is probably no option to check such improvement here.

---

### Official Review · Reviewer_cqFs · 2023-07-07

**Soundness:** 3 good
**Presentation:** 2 fair
**Contribution:** 4 excellent
**Rating:** 7
**Confidence:** 3

**Summary:**

This paper proposes an inverse reinforcement learning (IRL) algorithm for infinite horizon average reward Markov decision processes (AMDPs). At first, the authors show the stochastic policy mirror descent (SPMD) algorithm that achieves $\mathcal{O}(\varepsilon^{-1})$ rate of convergence. Then, the authors propose the inverse policy mirror descent (IPMD) algorithm that achieves $\mathcal{O}(\varepsilon^{-2})$ rate of convergence. The SPMD algorithm is compared with SAC on the MuJoCo benchmarks, and the SPMD achieves on-par performance with SAC. The IPMD algorithm is compared with f-IRL and IQL, and the experimental results show that IPMD is slightly better than f-IRL. Interestingly, the proposed algorithms perform much better than the baselines for the Humanoid environment that has many degrees of freedom.

**Strengths:**

- Originality: RL and IRL algorithms for AMDP with strong theoretical basis are novel, although Dvijotham and Todorov proposed IRL under the linearly solvable Markov decision processes (LMDP).
- Quality: The paper provides some strong theoretical analysis on AMDP.
- Clarity: The authors show detailed steps to prove the theorems in Appendix.
- Significance: The standard IRL algorithms deal with infinite horizon discounted reward problems under the assumption that the discount factor is known in advance. It is problematic because the expert's discount factor is usually unknown. The proposed method is promising because it can avoid tuning the discount factor.


**Weaknesses:**

- Some recent studies regarding AMDPs are not mentioned in the manuscript.


**Questions:**

Major comments
- Is the SPDM algorithm for AMDPs (Algorithm 1) an on-policy method? The objective function for training $\bar{\mathcal{Q}}^{\pi, \zeta}$ (35) suggests an on-policy algorithm, but I did not find the explanation. In addition, (35) does not use a target network. Is it a benefit of the property of the on-policy algorithm?
- I do not fully understand why $\nabla \omega (\pi)$ is usually unavailable, as described in Lines 143-144, because $\omega$ is selected by ourselves. Furthermore, it is unclear why its estimator is parameterized by $\phi$ because $\pi$ is parameterized by $\xi$ (33).
- Please show $g(\theta; \zeta)$ in detail because it is used to update $\theta$. For example, is $g(\theta_k; \zeta^\pi_k)$ the estimator of $\mathbb{E}_{(s, a)~d^\pi} [\nabla_\theta c(s, a; \theta)]$?
- The experimental results are promising, but one potential reason for successful results is that the reward function of the MuJoCo benchmark is well-shaped. For example, the reward of Walker2D is given by healthy_reward + forward_reward - control_cost: https://gymnasium.farama.org/environments/mujoco/walker2d/ . It may imply that the average reward formulation is more appropriate than the discount reward one in the MuJoCo benchmark. My interest is the performance of the proposed method when it is evaluated in sparse reward settings such as navigation tasks.

Minor comments
- The following paper proposes a framework based on stochastic mirror descent for AMDP. Please discuss the relationship between these papers and the proposed method.
  - Y. Jin and A. Sidford. (2020). Efficiently Solving MDPs with Stochastic Mirror Descent. In Proc. of ICML. G.
  - Neu et al. (2017). A unified view of entropy-regularized Markov decision processes. http://arxiv.org/abs/1705.07798 .
  - C.-Y. Wei, et al. (2020). Model-free Reinforcement Learning in Infinite-horizon Average-reward Markov Decision Processes. In Proc. of ICML.
- As mentioned in Strength, the following paper proposes a method to estimate the reward function under LMDP: K. Dvijotham and E. Torodov. (2010). Inverse Optimal Control with Linearly Solvable MDPs. In Proc. of ICML. It would be better to discuss their paper.


**Limitations:**

Minor comment: The authors briefly mention issues on potential societal impacts in A.10.

---

> ### Author Rebuttal · Authors · 2023-08-08
>
> Thank you for your time reviewing our paper. Your points are valid with respect to the arrangements of the paper. We appreciate your keen observation and suggestion regarding recent studies on Average-Reward Markov Decision Processes (AMDPs) that may contribute to our work. While we strive to provide a comprehensive review of relevant literature, we acknowledge that some recent studies might not have been explicitly referenced in the manuscript. We will conduct a thorough review of the literature to identify and appropriately reference any pertinent works that can enrich our discussion and provide context for our contributions.
>
> 1. The SPDM algorithm for AMDPs, as outlined in Algorithm 1, is designed to accommodate both on-policy and off-policy training modes. In practice, our RL algorithm implementation is based on off-policy training schemes with a target network, similar to the approach employed in SAC. However, the adaptability of SPDM allows for on-policy training as well.
>
> 2. We understand your query regarding the availability of $\nabla \omega(\pi)$. By unavailable we meant it needs stochastic estimation. While we can indeed choose $\omega$ ourselves, the precise value of $\omega(\pi)$ and its gradient $\nabla \omega(\pi)$ needs estimation, just as the differential $Q$-function. Notice that even though $\omega$ can be a general distance-generating function other than negative entropy, evaluating its value might require stochastic estimation from the sample we collected. Regarding the parameter $\phi$, we note that in cases where the action value function is parameterized separately from the policy, $\nabla \omega(\pi)$ might not share the same parameters as $\xi$ in Equation (33). $\omega$ may also be present in the form of a neural network or other models, which we denote its parameter $\phi$, the same as the $Q$-function. We recognize that this aspect could be clearer in our explanation and will revise the manuscript accordingly.
>
> 3. We appreciate your request for more detailed clarification regarding $g_k$. In practice, we employ a simple stochastic estimator using sample averages, denoted as $g(\theta; \zeta)= \tfrac{1}{N} \sum_{t=1}^N \nabla c (s_t,a_t;\theta)$, where $N=|\zeta|$ represents the number of samples in the collection $\zeta$. This estimator aids in the update of $\theta$ as part of the algorithm. In this sense, $g(\theta_k;\zeta_k)$ is indeed the estimator of $E_{(s,a)\sim d^{\pi_k}}[\nabla_{\theta} c(s,a;\theta_k)]$.
>
> 4. Your observation about the well-shaped reward functions in the MuJoCo benchmark is insightful. Indeed, the presence of dense rewards in MuJoCo tasks contributes to the success of our proposed method. We acknowledge your interest in the performance of our method in sparse reward settings, such as navigation tasks. This is an area of interest for us as well, and we plan to explore and discuss the behavior of our algorithm in such environments in future work.
>
> 5. Thank you for pointing out the relevance of [Jin and Sidford], [Neu et al. 2017], and [Chen et al., 2020] in the context of Mirror Descent-based methods.
> [Jin and Sidford] solve the average reward problem using linear programming-based methods, which is not comparable to policy gradient methods in practice due to scalability issues, although their analysis yields a novel complexity bound. [Neu et al. 2017] propose an entropy-regularized Reinforcement Learning framework, which covers important policy gradient methods and casts them as mirror descent or dual averaging. However, no convergence analysis is provided. [Chen et al., 2020] is more comparable to our RL analysis. However, the regret bound is known to be different from general convergence analysis. Nevertheless, all the above works consider MDPs with finite state and action spaces. Our work considers a more general setting where action and state can be continuous instead of categorical, for example, in robotics research and animal behavior studies.
>
> 6. We appreciate your reference to Dvijotham and Todorov's work on IRL under linearly solvable MDPs (LMDP). While their approach offers an effective control-oriented perspective on IRL, we recognize that using Maximum Likelihood Estimation may face limitations when applied to larger and more complex problems, such as those involving continuous state and action spaces. Our choice to formulate the problem under a maximum entropy framework accounts for these challenges and allows for a more robust and flexible approach. We will certainly expand on the discussion regarding the relationship between our approach and the LMDP framework in our forthcoming revision.
>
> Once again, we sincerely thank you for your thoughtful review, which has significantly contributed to the refinement of our paper. Please kindly refer to our global response for societal impacts.

---

### Official Review · Reviewer_9xxW · 2023-07-07

**Soundness:** 3 good
**Presentation:** 4 excellent
**Contribution:** 3 good
**Rating:** 6
**Confidence:** 4

**Summary:**

This paper aims to address the problem of inverse reinforcement learning (IRL) under the maximum entropy framework (MaxEnt-IRL) and an average reward criterion. The MaxEnt-IRL problem is formulated as a combination of an average reward Markov decision process (AMDP) and a dual IRL problem. The proposed algorithms, namely the SPMD algorithm and the IPMD algorithm, are utilized to solve the AMDP and dual IRL problem, respectively. The SPMD algorithm achieves a gradient computation step complexity of $\mathcal{O}(1/\epsilon)$ under general state and action spaces, while the IPMD algorithm has a complexity of $\mathcal{O}(1/\epsilon^2)$. The empirical studies conducted on the MuJoCo benchmark and various control tasks confirm the theoretical findings.


**Strengths:**

1. The idea of formulating the IRL problem as a dual IRL problem, incorporating an AMDP as a subproblem, is interesting.
2. The paper extends a previous RL algorithm [1] to general state and action spaces with a general function approximation class for AMDPs. The paper is clear and easy to follow.
3. The proposed algorithms are well-founded and supported by theoretical analyses of average reward Markov decision processes. The convergence analysis of the algorithms is provided under certain assumptions on the approximation function classes.

Overall, the paper is well-written and easy to follow. The proposed algorithms are well-motivated and supported by theoretical foundations. However, certain assumptions regarding function approximation classes and the critic step may be too restrictive, potentially resulting in the failure of convergence in scenarios involving deep neural networks. Despite these concerns, the theoretical contributions of the paper could be of interest to the community, and I recommend the acceptance of the paper.

[1] Tianjiao Li, Feiyang Wu, and Guanghui Lan. Stochastic first-order methods for average-reward markov decision processes.arXiv preprint arXiv: 2205.05800, 2022.


**Weaknesses:**

1/ The advantages of employing the average reward criterion over discounted IRL are not adequately explained in the paper.
2/ Assumption 3.2 assumes weak convexity and Lipschitz continuity for the approximated Q-functions. These assumptions may not hold in practice, particularly when neural networks are used for function approximation. As the global convergence result of SPMD relies on the convexity of the approximated Q-function and the regularizer h, it may not hold in practical scenarios.
3/ Assumption 3.6 imposes a restriction that the errors of the critic step, typically represented by neural networks, are bounded. This assumption may be overly restrictive, as estimations can be highly inaccurate in many cases. The convergence result for general function approximation classes may also fail to hold in practice.
4/ When the reward function could be parameterized by neural networks, which is normally the case in practice, Assumption 4.1 would be too strong.
5/ In the reward recovery experiment, it would be beneficial to compare the proposed algorithms with discounted IRL algorithms since the expert is trained in a discounted environment (the author trained the expert agent using SAC with a discount factor of 0.99).
6/ Including the running time of the proposed algorithms and the baselines would provide valuable insights.


**Questions:**

Seen in the weaknesses section.

**Limitations:**

Seen in the weaknesses section.

---

> ### Author Rebuttal · Authors · 2023-08-08
>
> We greatly appreciate your thoughtful review and the points you've raised. Your insights have been invaluable in refining our paper. Below, we address each of your comments:
>
> 1. We apologize for any lack of clarity in explaining the advantages of employing the average reward criterion over discounted IRL. We appreciate your comprehensive understanding of this aspect. Discounted IRL has challenges posed by guessing the discount factor. Our average-reward criterion formulation addresses this issue effectively. First, considering demonstrations using an average-reward metric, our method excels in both reward and policy recovery. Furthermore, when the demonstration indeed uses a discount factor, but the discount factor is unknown, our approach yields improved reward estimates without the need for manual discount factor guesswork. We will enhance our paper to elucidate these advantages more explicitly.
>
> 2. We thank you for highlighting the potential limitations of Assumption 3.2, particularly in the context of neural network-based function approximation. You're correct in noting that neural networks don't guarantee Lipschitz continuity. However, as we assume the state and action space are bounded sets, it is reasonable to assume there exists a Lipschitz constant, which can be fairly large. Regarding the convexity of the approximated Q-function, we have additional analysis when the approximated Q-function plus the regularizer h is not convex, which leads to SPMD converging to a stationary point. See Theorem 3.7. Please refer to our overall response for a detailed discussion.
>
> 3. Your observation about the potentially restrictive nature of Assumption 3.6 is insightful. We recognize that bounded errors may not accurately capture all cases, especially when dealing with neural network-based estimations. Your suggestion to consider relaxing this assumption aligns with our intention to make our approach more applicable to a broader range of scenarios. We will explore ways to alleviate this assumption while ensuring the reliability of our convergence results. Please refer to our overall response for a detailed discussion.
>
> 4. We appreciate your insight regarding the applicability of Assumption 4.1 when dealing with neural network parameterized reward functions. You're right that neural networks don't inherently guarantee Lipschitz continuity or bounded gradients. But as argued above, since the state space and the action space are compact (bounded and closed), the parameterized reward function is indeed Lipschitz continuous as long as its gradient is bounded, which is much less restrictive. It is even less restrictive when simpler models are preferred. For instance, the assumption aligns well with linear reward functions over feature spaces, commonly used in animal behavior studies, which brings room for further inference and interpretation. We will enhance the clarity of this rationale in our manuscript.
>
> 5. Both IQL and f-IRL are trained with the same discount factor of 0.99 in all experiments, the same as the expert demonstrations.
>
> 6. We thank you for highlighting the value of including running time information for our proposed algorithms and baselines. Our RL algorithm has the same computation efforts as similar algorithms in stable-baselines like PPO, SAC, etc. Training an agent in a single-thread environment with 5 million steps typically takes around 3.5 hours on the Apple M1 laptop. Our IRL algorithms, although carries additional reward estimation in each step, do not impose too much computational burden. Training an agent in a single-thread environment with 5 million steps takes around 3.5 to 4 hours. Please refer to our overall response for all running times. Notice that Humanoid is a larger instance and thus takes longer to train.
>
> Once again, we extend our gratitude for your thorough review and constructive feedback, which have significantly contributed to improving our paper.

---

> > ### Comment · Reviewer_9xxW · 2023-08-15
> > **Thank you for the clarification!**
> >
> > Thank you for the clarification!

---

### Author Rebuttal · Authors · 2023-08-08

1. Regarding assumptions being too restrictive:
We acknowledge that some assumptions made in the paper are sometimes too restrictive. However, our particular problem setting has a special implication for the assumptions we made.  Assumption 2.1 (uniform ergodicity) is considered restrictive but often necessary for analysis, albeit potential violations in practice. Assumption 3.2's practicality hinges on model choices; if neural networks are employed, gradient "clipping" may be utilized. This also applies to Assumptions 4.1 and 4.2. Assumptions 3.3, 3.6, and 4.3, concerning stochastic estimator proximity, are less restrictive given bounded state and action spaces.
There are possible remedies to encourage our algorithm to satisfy those assumptions. First, we can "clip" large gradients by a constant; or add penalty terms regarding the norm of the gradient. Regarding the span semi-norm contraction property, we can construct a $J-$step operator in the policy evaluation step, which is well explored in the literature (see [Puterman 1994], [Zhang 2021]).

Puterman, M. L. (2014). Markov decision processes: discrete stochastic dynamic programming. John Wiley & Sons.

Zhang, S., Zhang, Z., Maguluri, S. T. (2021). Finite Sample Analysis of Average-Reward TD Learning and $ Q $-Learning. Advances in Neural Information Processing Systems, 34, 1230-1242.

In practice, it is usually obvious when things go wrong, e.g., when the differential Q-function explodes to 10e6. This can happen due to rare occasions, and the culprit is usually detectable, e.g., a large learning rate.

2. Hyperparameter settings:
The policy network employs two fully connected hidden layers of dimension 256 each, taking actions as input and outputting a distribution. Both the Q network and reward function share the same architecture, with ReLU activation used in hidden layers. A double Q-learning technique minimizes overestimation.
During training, we found that setting the entropy coefficient term to 0.01 make training stable and efficient. The learning rate is set to be 3e-4. Each step of the algorithm samples 512 state-action sample pairs. Additionally, we include our running time of APMD and IPMD for all MuJoCo tasks in the pdf file attached.

3. Experiments analysis:
We acknowledge the request for a deeper discussion of algorithmic improvements and deficiencies in various environments. While further performance analysis and tuning are indeed possible, our primary focus was on theoretical foundations. One possible reason Ant is falling behind is that Ant has more ground contact since it has more legs. This will impact the ergodicity of the MDP and our assumption on the 1-step contraction. It is harder to transition from an arbitrary state to another arbitrary state as it involves multiple legs working together. One remedy is that we construct a $J$-step contractive operator for policy evaluation, as done in [Chen 2021]. For the success of humanoid, it is possibly due to a slightly different policy evaluation scheme where the entropy term from the policy no longer plays a part, as the term $r-\rho$ cancels out the additional regularization and entropy of the policy. This is different from discounted setting. We suspect this brings more stable training and thus higher performance.

4. Ethical discussion:
Our focus is on methodology, and we acknowledge that IRL, like many other machine learning techniques, has potential implications if misused. IRL can be used in violation of privacy (as the reviewer mentioned) by inferring an individual's intentions and preferences, potentially crafting convincing social engineering attacks or phishing attempts; IRL can also be used to model the behavior of specific demographics, which could result in biased algorithmic decision-making, leading to unfair treatment or discrimination against certain groups, etc.

---

### Decision · Program_Chairs · 2023-09-21

**Decision:**

Accept (poster)

**Comment:**

The reviews for this paper were mixed, with some reviewers accepting it and some reviewers rejecting it, giving the paper an average borderline score. The paper considers the intersection of average reward & inverse RL which is giving it a relatively narrow scope. After reading the paper and discussing related work with the authors, the novelty of the paper is now clear to me, but remains somewhat limited given the already narrow scope and existing related work. Nevertheless, the paper makes interesting contributions that can later on be generalised to other settings. I encourage the authors to include a discussion on how their work can be used more broadly, for example, for other problems that involve average reward or more general utilities. I also encourage them to include the detailed response and comparisons they have provided in the rebuttal regarding the related work. After further considerations, I am recommending to accept the paper.